# Graph-based pan-genome reveals structural and sequence variations related to agronomic traits and domestication in cucumber

Hongbo Li[1,2,3,9], Shenhao Wang[4,9], Sen Chai[2], Zhiquan Yang [5], Qiqi Zhang[1], Hongjia Xin[1], Yuanchao Xu[1], Shengnan Lin[1,6], Xinxiu Chen[2], Zhiwang Yao[2], Qingyong Yang [5], Zhangjun Fei [7,8], Sanwen Huang [3] & Zhonghua Zhang [2✉]

Structural variants (SVs) represent a major source of genetic diversity and are related to numerous agronomic traits and evolutionary events; however, their comprehensive identification and characterization in cucumber (*Cucumis sativus* L.) have been hindered by the lack of a high-quality pan-genome. Here, we report a graph-based cucumber pan-genome by analyzing twelve chromosome-scale genome assemblies. Genotyping of seven large chromosomal rearrangements based on the pan-genome provides useful information for use of wild accessions in breeding and genetic studies. A total of ~4.3 million genetic variants including 56,214 SVs are identified leveraging the chromosome-level assemblies. The pan-genome graph integrating both variant information and reference genome sequences aids the identification of SVs associated with agronomic traits, including warty fruits, flowering times and root growth, and enhances the understanding of cucumber trait evolution. The graph-based cucumber pan-genome and the identified genetic variants provide rich resources for future biological research and genomics-assisted breeding.

[1] Key Laboratory of Biology and Genetic Improvement of Horticultural Crops of the Ministry of Agriculture, Sino-Dutch Joint Laboratory of Horticultural Genomics, Institute of Vegetables and Flowers, Chinese Academy of Agricultural Sciences, 100081 Beijing, China. [2] Engineering Laboratory of Genetic Improvement of Horticultural Crops of Shandong Province, College of Horticulture, Qingdao Agricultural University, 266109 Qingdao, China. [3] Shenzhen Branch, Guangdong Laboratory for Lingnan Modern Agriculture, Shenzhen Key Laboratory of Agricultural Synthetic Biology, Genome Analysis Laboratory of the Ministry of Agriculture and Rural Affairs, Agricultural Genomics Institute at Shenzhen, Chinese Academy of Agricultural Sciences, 518120 Shenzhen, China. [4] College of Horticulture, Northwest A&F University, 712100 Yangling, Shanxi, China. [5] College of Informatics, Huazhong Agricultural University, 430070 Wuhan, China. [6] Key Laboratory of Horticultural Plant Biology, College of Horticulture and Forestry Sciences, Huazhong Agricultural University, 430070 Wuhan, China. [7] Boyce Thompson Institute, Cornell University, Ithaca, NY 14853, USA. [8] US Department of Agriculture-Agricultural Research Service, Robert W. Holley Center for Agriculture and Health, Ithaca, NY 14853, USA. [9]These authors contributed equally: Hongbo Li, Shenhao Wang. ✉email: zhangzhonghua@qau.edu.cn

With the advance of high-throughput sequencing technologies, an increasing number of studies have characterized genetic variants including single-nucleotide polymorphisms (SNPs) and small insertions/deletions (InDels) through mapping short reads to a single reference genome[1–4]. However, recent reports have suggested the limitation of this approach in capturing the full spectrum of genetic diversity, especially in identifying structural variants (SVs)[5,6]. Since numerous studies have indicated that some genes controlling agronomic traits are absent in the widely used reference genomes due to presence/absence variations (PAVs)[7,8] and SVs play critical roles in genome evolution and genetic control of agronomical traits in plants[9–11], it is necessary to develop multiple representative reference genomes to facilitate SV detection and characterization. Recent pan-genome studies in human and plant species have uncovered species-wide biodiversity with an emphasis on the characterization of SVs[12–16]. Additionally, increasing studies have focused on the construction of graph-based genomes in which loci with common variants are represented by alternative sequences. Therefore, graph-based genomes contain not only reference sequences but also variants in a population, providing a promising approach for pan-genome representation[17–19].

Cucumber (*Cucumis sativus* L.) is one of the major vegetable crops and also serves as a model system for plant sex determination and vascular development research[20]. Based on the reference genome of the 'Chinese long' inbred line 9930 (ref. [20]), several population genomic studies utilizing Illumina short reads have reported the systematic characterization of genetic variants such as SNPs and small InDels, and further revealed the domestication history and divergence of cucumber[21,22]. A previously reported resequencing-based cucumber SV map revealed a copy number variation that defines the *Female* (*F*) locus[23], highlighting the role of SVs in favorable trait determination. However, technical limitations and the relatively low quality and few numbers of reference genomes available at that time hampered the identification of genetic variants especially SVs among cucumber individuals, thus emphasizing the necessity to develop additional reference genomes from more diverse accessions.

In this study, we construct chromosome-scale assemblies for an additional 11 representative accessions comprising three wild and eight cultivated cucumbers. Together with the 9930 genome[24], we build a graph-based pan-genome and detect millions of genetic variants including numerous SVs and genotype them in a 115-line core collection[21], empowering the identification of variants potentially associated with agronomic traits and domestication. Our graph-based pan-genome and the genetic variants identified herein will accelerate variome-based breeding in cucumber.

## Results
**Chromosome-scale assemblies of 11 cucumber accessions**. We selected 11 representative accessions from the 115-line core collection based on their reported phylogenetic relationship[21] (Supplementary Fig. 1). They comprised two East-Asian lines (XTMC and Cu2), three Eurasian lines (Cuc37, Gy14, and 9110gt), one Xishuangbanna line (Cuc80), and five Indian lines (Cuc64, W4, W8, Hx14, and Hx117). Among them, Cuc64, W4, and W8 belong to *C. sativus* var. *hardwickii*, which is suggested to be the wild progenitor of cultivated cucumber[21]. We compared genetic distance measures (Modified Rogers distance, *MR* and Cavalli-Sforza and Edwards distance, *CE*) and community diversity index (Shannon's diversity index, *SH*) between the 11 accessions plus the 9930 reference (a total of 12 accessions) and the 115 lines, and found that the 12 accessions had larger *MR* (0.495 versus 0.377) and *CE* (0.498 versus 0.377) while a slightly

reduced *SH* (9.444 versus 9.510) compared with the 115 lines (Supplementary Table 1), indicating that these 12 accessions are genetically diverse. We also calculated the genetic coverage value and found that ~84% of genetic diversity in the 115-line population was captured by the 12 accessions (Supplementary Table 1), indicating the representativeness of these accessions.

We performed PacBio sequencing for the 11 accessions and generated 15.9–22.4 Gb of data, representing 45.4–64.0× genome coverages (Supplementary Table 2). Chromosome-scale genome assemblies were generated based on PacBio reads together with other data from multiple sequencing platforms. For three accessions, Cuc37, Cuc80, and Cuc64, PacBio reads were assembled into contigs and then polished using previously generated Illumina reads (Supplementary Table 3). We then split potential mis-assembled contigs showing obvious inconsistency with genetic markers from four linkage maps[25–28] (Supplementary Fig. 2). The assembled contigs were next connected into scaffolds based on 10X Genomics data (Supplementary Table 4), and Hi-C data of ~200× coverage (Supplementary Table 5) were used to order and orientate the scaffolds into seven pseudo-chromosomes. The high consistency between assembled sequences of Cuc37, Cuc80, and Cuc64 and the corresponding Hi-C data indicate that these genome assemblies are of high accuracy (Supplementary Fig. 3). For the eight other accessions, contigs were also assembled using PacBio reads, polished with Illumina reads, and then clustered into seven pseudo-chromosomes on the basis of four reported genetic maps[25–28]. A total of 830–1015 final contigs with N50 ranging from 1.7 to 5.3 Mb were generated (Table 1 and Supplementary Table 6). These assemblies had total lengths ranging from 232.5 to 251.1 Mb with BUSCO[29] scores ranging between 96.4 and 97.7%, indicating their high completeness within genic regions.

A total of 32.5–38.5% of these genomes were predicted to be transposable elements, with long terminal repeat retrotransposons (LTR-RTs) being the most abundant class (14.3–19.3%) (Table 1 and Supplementary Table 7). We further identified 324–531 intact LTR-RTs and classified them into 453 families (Supplementary Table 8 and Supplementary Data 1). Prediction of protein-coding genes was performed for each of the 11 genomes, resulting in 24,490–26,033 genes with average gene lengths ranging from 3182 to 3302 bp and CDS lengths between 1075 and 1124 bp (Table 1 and Supplementary Table 9). The mean BUSCO[29] score of the predicted genes was estimated to be around 96.0%, suggesting that the predicted gene models in these genomes were sufficient for downstream analyses (Table 1).

**Karyotype evolution of cucumber**. A previous study reported six chromosomal rearrangements (inversions) between a wild (Cuc64) and a cultivated cucumber (Gy14) based on fluorescence in situ hybridization (FISH)[26], but the coordinates of these inversions on genomes and their presence/absence nature among other cucumbers remain unexplored. Here, we confirmed the presence of these six large inversions by aligning the chromosome-level genome assemblies of Cuc64 and 9930, and identified an additional megabase-scale inversion, which might be overlooked due to the low density of FISH markers used in the previous study. Mapping Hi-C data of Cuc64 to the 9930 genome showed strong interaction signals around breakpoints of all these inversions, highlighting the advantage of Hi-C in the identification of large-scale chromosomal rearrangements (Fig. 1a–c and Supplementary Table 10). We found that in addition to Cuc64, W8, another wild cucumber, also carried the three inversions on chromosome 5 (Fig. 1d, e), as clearly evidenced by alignments around breakpoints in the three corresponding contigs (Supplementary Fig. 4), while the wild cucumber W4 did not contain any

**Table 1 Summary statistics of cucumber genome assemblies.**

| Accession | Group | Total length (Mb) | No. of contigs | Contig N50[a] (Mb) | Completeness[b] (%) | No. of genes | Completeness[b] (%) | TE content (%) |
|---|---|---|---|---|---|---|---|---|
| 9930[c] | East-Asian | 224.8 | 174 | 8.9 | 97.7 | 24,714 | 95.5 | 32.5 |
| XTMC | East-Asian | 240.1 | 926 | 2.1 | 97.0 | 25,167 | 96.6 | 37.2 |
| Cu2 | East-Asian | 247.1 | 851 | 5.3 | 97.5 | 25,382 | 97.6 | 37.7 |
| Cuc37 | Eurasian | 238.4 | 967 | 3.8 | 97.4 | 24,490 | 94.0 | 37.2 |
| Gy14 | Eurasian | 239.4 | 926 | 2.1 | 97.4 | 25,042 | 95.6 | 35.2 |
| 9110gt | Eurasian | 242.9 | 830 | 3.9 | 97.2 | 24,992 | 96.5 | 38.5 |
| Cuc80 | Xishuangbanna | 237.4 | 923 | 4.3 | 97.7 | 24,578 | 96.4 | 36.1 |
| Cuc64 | Indian | 232.5 | 842 | 4.4 | 97.7 | 24,583 | 97.0 | 35.4 |
| W4 | Indian | 251.1 | 894 | 4.7 | 97.3 | 25,703 | 96.6 | 38.0 |
| W8 | Indian | 241.9 | 907 | 4.3 | 97.2 | 25,531 | 95.6 | 36.7 |
| Hx14 | Indian | 234.6 | 865 | 1.7 | 97.3 | 24,914 | 96.8 | 36.4 |
| Hx117 | Indian | 243.7 | 1,015 | 2.1 | 96.4 | 26,033 | 93.3 | 36.4 |

[a]N50 refers to the size above which half of the total length of the sequence set can be found.
[b]Based on BUSCO assessment.
[c]Published reference genome of cucumber '9930'[24].

of these inversions (Fig. 1e). Based on the phylogeny of the 12 accessions (Fig. 1d), we can deduce that the inversions on chromosomes 4 and 7 could have occurred after the divergence between W8 and Cuc64. Considering that all cultivated and some wild accessions do not possess any inversions on chromosome 5, we propose that these inversions could have occurred during the evolution of wild cucumbers. The breakpoints and presence/absence information of these seven chromosomal rearrangements among the 12 accessions provide additional insights into cucumber karyotype evolution. Since large segmental inversions can lead to recombination suppression in these regions[20], our inversion map provides a guide for properly selecting parental lines to construct segregating populations between wild and cultivated cucumbers.

**Pan-genome of cucumber**. We constructed a protein-coding gene-based pan-genome for cucumber by clustering 299,692 predicted gene models from the 11 accessions and 9930, which resulted in 26,822 non-redundant pan-gene clusters comprising 18,651 core (genes in these clusters present in all 12 accessions) and 8171 dispensable clusters (genes in these clusters absent in at least one accession) (Supplementary Table 11 and Supplementary Data 2). Core genes occupied over 80% of the total genes in each accession (Fig. 2a). Simulation of the pan-genome size indicated that the curve of the number of gene clusters nearly reached a plateau when the number of accessions was greater than or equal to nine (Supplementary Fig. 5a). We also observed that the number of added gene clusters declined rapidly, and only 98 additional clusters were detected when adding the 12th accession (Supplementary Fig. 5b), further indicating the representativeness of the 12 cucumber accessions sampled in this study. Gene ontology (GO) term analysis indicated that core genes were enriched for several essential biological processes including macromolecule modification, glycosylation, and phosphorus metabolic process, while biological processes such as DNA integration, response to several endogenous factors including auxin and other hormones, and telomere maintenance were enriched in dispensable genes (Fig. 2b, c, Supplementary Tables 12 and 13). In addition, dispensable genes were shorter than core genes ($p < 2.2e-16$, Student's $t$-test), and were expressed at a significantly lower level ($p < 2.2e-16$, Wilcoxon rank-sum test) (Fig. 2d, e). The significantly lower non-synonymous/synonymous substitution ratio ($K_a/K_s$) of core genes than that of dispensable genes ($p < 2.2e-16$, Wilcoxon rank-sum test) suggests that dispensable genes may have undergone less stringent

purifying selection (Fig. 2f). Together these results indicate that dispensable genes exhibit faster evolution rates and may display diverse functions in cucumber adaptation to particular environmental conditions.

**Genetic variation and the variant-integrated graph-based pan-genome**. Our high-contiguity genome assemblies allowed accurate identification of genetic variants in terms of SNPs, small InDels (<50 bp in size), and SVs (large InDels, inversions, and translocations, ≥50 bp in size) through alignments of inter-genomic collinear blocks (Fig. 3a–c). We identified 2,902,954 SNPs and 1,388,197 small InDels using the 9930 genome as the reference. Of these variants, about 2.5% of SNPs and 1.5% of small InDels caused changes of start/stop codons, splicing sites, encoded amino acids, or frameshifts, which may contribute to the diversity of gene functions (Fig. 3d, e and Supplementary Tables 14-16). We classified large InDels into two categories: canonical InDels with exact breakpoints flanked by well-aligned collinear regions, and complex InDels that are pairs of unaligned sequences of different lengths within regions in good collinearity. A total of 53,912 large InDels comprising 17,130 canonical insertions, 19,334 canonical deletions, 9399 complex insertions, and 8049 complex deletions related to the 9930 reference genome were identified (Supplementary Tables 17-20 and Supplementary Data 3). These SVs exhibited a significantly lower density around centromeres (Fig. 3f, g). Moreover, we detected 196 inversions (<1 Mb in size) representing a total of 5.47 Mb of sequences and 2106 translocations compared with the 9930 reference genome (Supplementary Tables 21-23 and Supplementary Data 3). To evaluate the accuracy of the identified SVs, we integrated several state-of-the-art resequencing-based SV calling methods consisting of read depth (RD), split-read (SR), and read-pair (RP) analyses to examine how many SVs can be supported by Illumina sequences. We found that 69.7–85.5% canonical deletions, 89.5–96.1% of canonical insertions, 62.9–86.5% of complex deletions, and 83.6–91.7% of complex insertions could be supported by either RD, SR, or RP evidence (Supplementary Tables 24-27), and a proportion of the remaining SVs might only be detected using long reads (Supplementary Fig. 6), suggesting that our SV dataset is of relatively high confidence.

To expand our knowledge of SVs from the 12 accessions to the population scale, we constructed a graph-based pan-genome by integrating the sequences and coordinates of identified SVs into the 9930 linear reference genome sequences (Fig. 3h, i). The graph structure enabled proper mapping of short reads derived

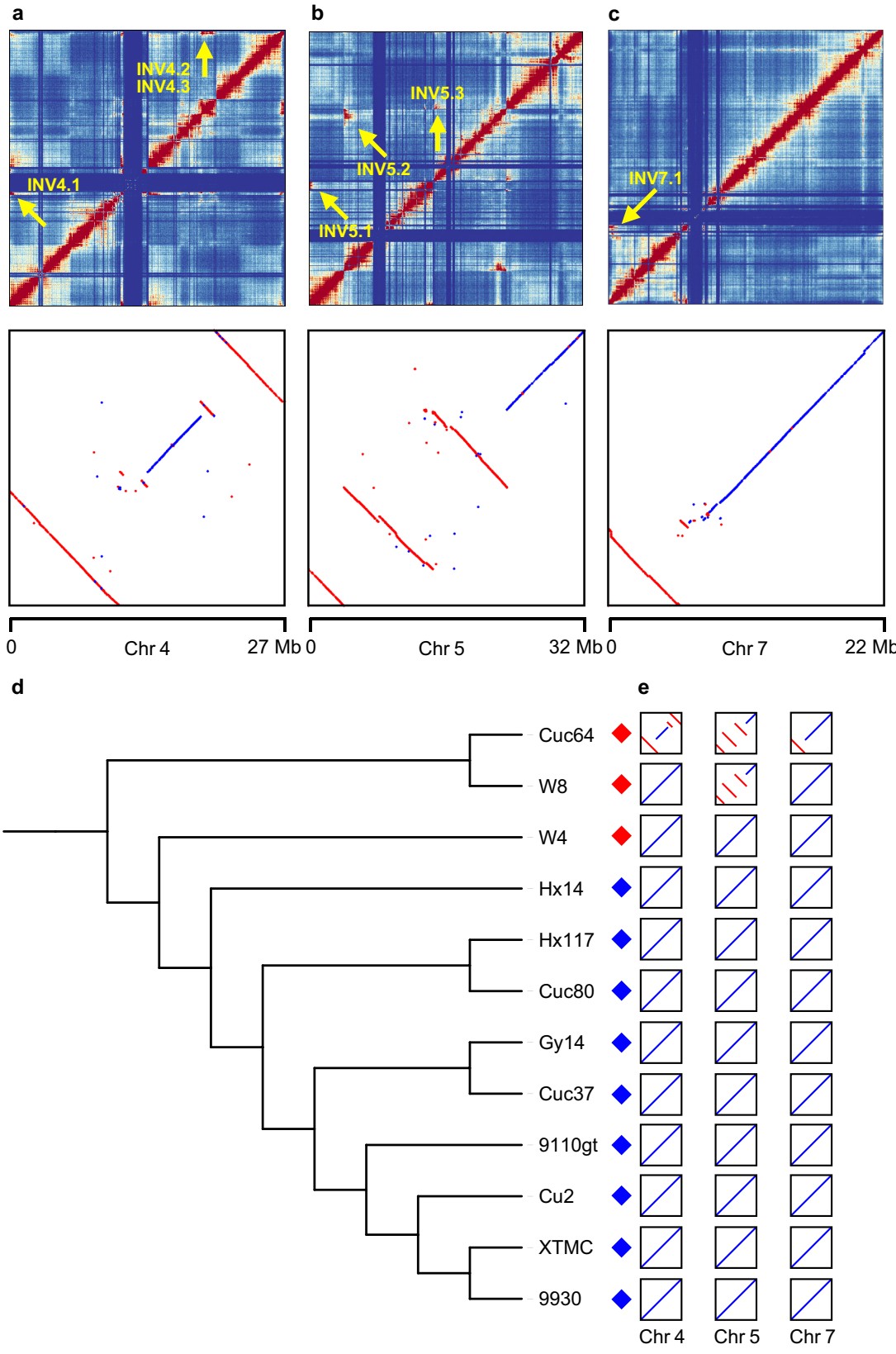

from SV flanking regions by embedding both reference and alternative allele sequences, thereby facilitating SV genotyping in the 115-line cucumber population[21]. We next performed genome-wide association studies (GWAS) using these genotyped SVs for three traits: female flower rate on a primary branch, fruit spine/wart density, and branch number. Two reported genes involved in cucumber sex determination ($m$[30] and $F$[31,32]) were detected in our association signals (Supplementary Fig. 7a), and the highest peak of fruit spine/wart density on chromosome 6 directly pinpointed the reported causative SV

**Fig. 1 Karyotype evolution of cucumber. a–c** Megabase-scale chromosomal rearrangements between genomes of Cuc64 and the 9930 reference on chromosome 4 (**a**), 5 (**b**), and 7 (**c**). Chromatin interaction heatmaps at the 50 kb resolution when mapping Hi-C reads of Cuc64 to the 9930 reference are shown in the upper panel. Red and blue colors indicate strong and weak interactions, respectively. Inversions are marked by yellow arrows. Genome alignment dot plots are displayed in the lower panel (x-axis: 9930, y-axis: Cuc64). **d** Phylogenetic relationships among the 12 cucumber accessions. Red and blue diamonds indicate wild and cultivated cucumbers, respectively. **e** Schematic diagrams of the distribution of large-scale chromosomal rearrangements among the 12 cucumber genomes. Red lines represent the inverted segments relative to 9930. Source data are provided as a Source Data file.

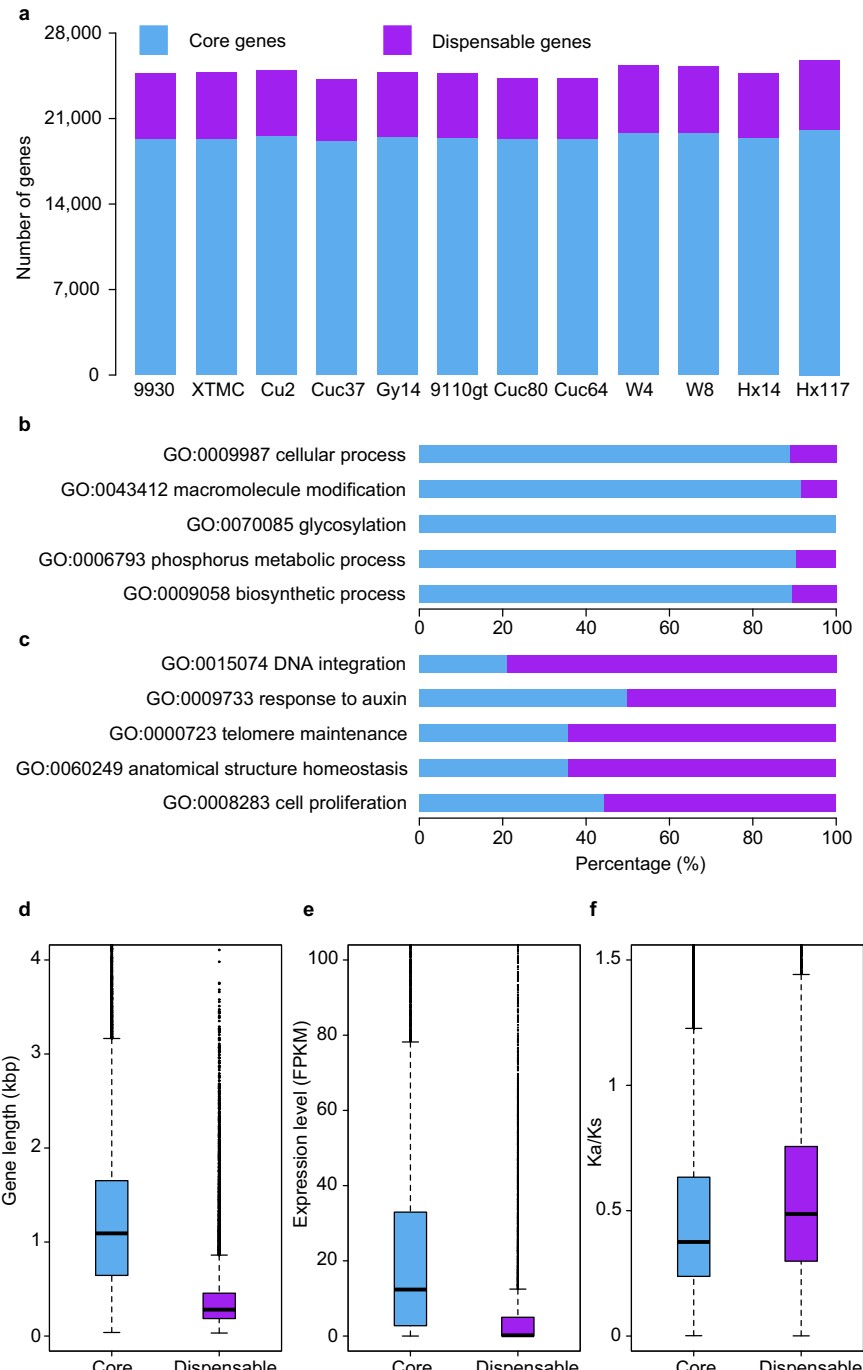

**Fig. 2 Pan-genome of cucumber. a** Number of core and dispensable genes in 12 cucumber accessions. **b, c** Top five GO terms (biological processes) enriched in core (**b**) and dispensable (**c**) genes. **d–f** Boxplots showing the median and upper and lower quartiles of gene length (**d**), expression level (**e**), and non-synonymous/synonymous substitution ratio ($K_a/K_s$) (**f**) of core and dispensable genes. The whiskers extend to 1.5 times of the interquartile range. Numbers of core and dispensable genes in **d** and **e** are 18,636 and 8396, respectively. Number of gene pairs used to compute $K_a/K_s$ in **f** is 701,248 for core genes and 66,879 for dispensable genes. Source data are provided as a Source Data file.

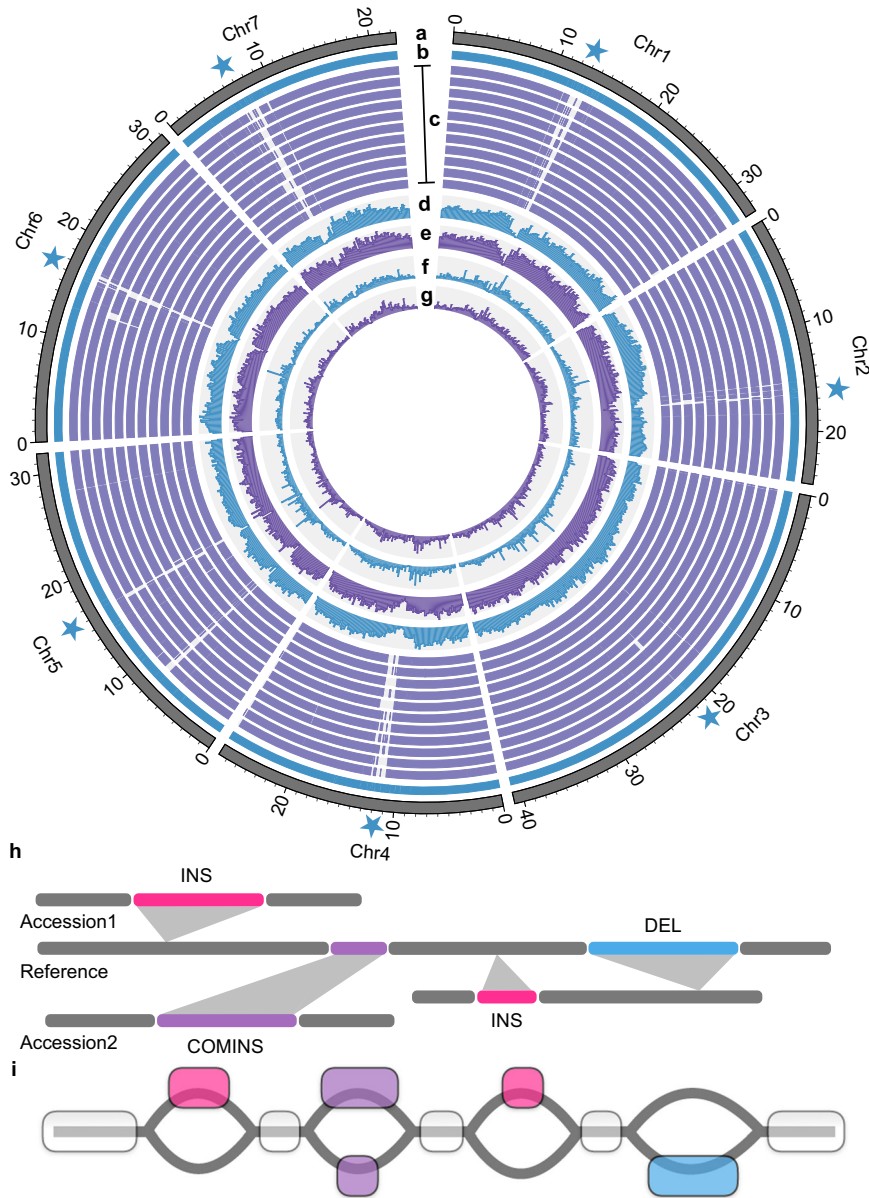

**Fig. 3 Collinear alignment blocks, genetic variants, and graph-based pan-genome of cucumber. a** Pseudo-chromosomes of the 9930 reference. Pentagrams mark approximate positions of centromeres. **b** Contig tracks of the 9930 reference. Gaps are shown by white bars. **c** Collinear alignment blocks of genomes of XTMC, Cu2, Cuc37, Gy14, 9110gt, Cuc80, Cuc64, W4, W8, Hx14, Hx117 (from upper to lower) to the 9930 reference. White bars represent non-synteny regions. **d, e** Variant density histograms of SNPs (**d**) and small InDels (**e**) (number/200 kb; 0–5000 for SNPs and 0–2000 for small InDels). **f, g** Distribution histograms of large insertions (**f**) and deletions (**g**) (number/200 kb; 0–120 for large insertions and deletions). **h** Schematic illustration of SVs from genomes of two example accessions and the linear reference genome sequences used to construct the graph-based pan-genome. Colored bars represent SV sequences and gray bars denote reference-type fragments. INS insertion, DEL deletion, COMINS complex insertion. **i** Cucumber graph-based pan-genome that integrates sequences and positions of SVs while preserves the linear reference coordinates. Transparent blocks illustrate sequences conserved among genomes of the 11 accessions and the reference, while colored blocks show fragments with SVs compared to the reference genome. Gray lines denote possible paths within the graph.

(SV_COMINS_6G012210) upstream of *CsGL3* (ref. [33]) (Supplementary Fig. 7b). We identified a 59 bp canonical insertion (SV_INS_7G011800) that was strongly associated with branch number, which has not been reported before (Supplementary Fig. 7c). This insertion was localized 2593 bp upstream of *Csa9930_7G025850* encoding an *Arabidopsis* BYPASS1 homolog that controls plant architecture by regulating root-derived signal production[34]. These results indicate that our graph-based pan-genome can act as a valuable platform to genotype SVs in large populations and deploy SV-based association analyses of agronomically important traits in cucumber.

**Functional impact of structural variants**. SVs usually contribute to phenotype variations to a large extent[35,36]. In order to associate potential SVs with functionally important genes, we screened our SV dataset and found 2624 SVs affecting CDS of 2,712 predicted genes (Supplementary Tables 17-23 and Supplementary Data 4).

Cucumber fruits display a rich diversity of spines and warts[37]. Several genes responsible for this trait have been isolated, acting as a complex regulatory network of the initiation and development of fruit spines and warts[38,39]. Considering the high diversity of fruit spines and warts in cucumber (Supplementary Fig. 8), we investigated allelic variation patterns for six previously

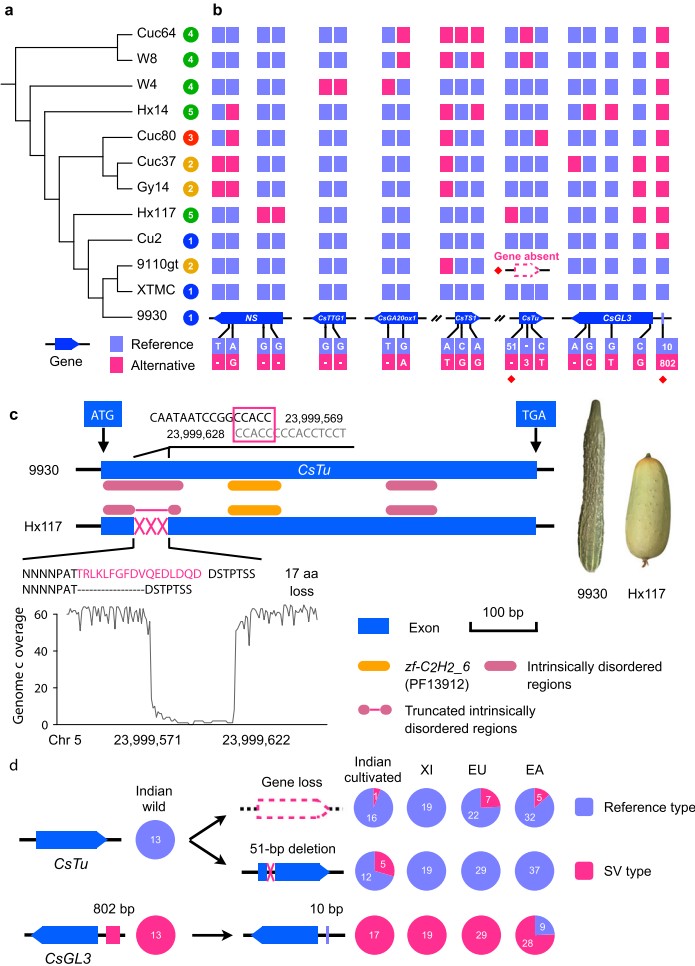

**Fig. 4 Allelic variants in genes involved in cucumber fruit spine and wart development. a** Phylogenetic relationships of 12 cucumber lines constructed using SNPs inside six genes related to fruit spine and wart development. Numbers 1–5 denote accessions belonging to the East-Asian, Eurasian, Xishuangbanna, Indian wild and Indian cultivated groups, respectively. **b** Haplotype information of three SVs, nine SNPs, and eight small InDels within the six genes among the 12 accessions. SVs are marked by red diamonds. **c** The 51 bp deletion in accession Hx117 causing the loss of 17 amino acids in CsTu. A 5 bp microhomolog (CCACC; red box) around the breakpoint of the deletion is shown. Coverage in the 50 bp left- and right-flanking regions of this deletion when mapping PacBio reads of Hx117 to the 9930 reference genome is depicted. Phenotypic variance of fruit spines and warts in 9930 and Hx117 is also shown. **d** Differential selection of *CsTu* and *CsGL3* alleles in cultivated cucumbers. Pie charts indicate the number of accessions in corresponding cultivated cucumber groups carrying reference or non-reference SV alleles. Source data are provided as a Source Data file.

characterized genes (*NS, CsTTG1, CsGA20ox1, CsTu, CsTS1,* and *CsGL3*)[33,37,40–43]. We identified 20 likely functionally important variants localized within or close to these genes, comprising three SVs, nine non-synonymous substitutions, and eight frameshift/in-frame InDels in the 12 accessions (Fig. 4b). Phylogenetic analysis using SNPs within these six genes suggested that the two Indian accessions (Cuc64 and W8), whose fruits showed small and low-density spines and warts, possessed the ancestral state of these loci (Fig. 4a and Supplementary Fig. 8). Regarding the three SVs, we identified a 51 bp canonical deletion in Hx117 (SV_DEL_5G021710) localized at the CDS of *CsTu* that encodes a $C_2H_2$ zinc-finger transcription factor, leading to the loss of 17 amino acids within a predicted intrinsically disordered region (Fig. 4c). This region presents putative binding functions[44,45]; thus, the deletion might disrupt the normal function of CsTu. We further examined the flanking sequences of this deletion and found a 5 bp microhomology (CCACC) around each side of breakpoints, suggesting that microhomology-mediated break-induced replication (MMBIR) likely gave rise to this deletion (Fig. 4c). The two other SVs were reported previously:

SV_DEL_5G021700, a 4895 bp canonical deletion leading to the complete loss of *CsTu*[37], and SV_COMINS_6G012210, a 10 bp sequence replaced by an 802 bp complex insertion upstream of *CsGL3* (ref. [33]). Leveraging the graph-based pan-genome, we genotyped these three SVs in the 115-line cucumber population[21] and found that the 4895 bp deletion was present in one Indian cultivated, seven Eurasian and five East-Asian accessions, while the 51 bp deletion was only found in five of the Indian cultivated cucumbers (Fig. 4d). Loss of *CsTu* due to the 4895 bp deletion confers cucumber fruits non-warty[37], which is preferred and has been selected in some regions of Europe, America, and Asia. The 10 bp/802 bp substitution upstream of *CsGL3* possessed a group-specific feature: the 10 bp upstream allele was only present in nine East-Asian accessions (Fig. 4d). Compared with accessions harboring the 802 bp upstream allele, accessions carrying the 10 bp upstream allele displayed a significantly elevated fruit spine density[33], which is indeed favored in several Asian countries. This could be the reason that this allele has been selected in some East-Asian cultivars. Our analyses identified an SV that affected the CDS of *CsTu*, which is worthy of further functional

characterization. These results also reveal the selection landscape of SVs in functionally important genes involved in the regulatory network of cucumber fruit spine and wart development.

The wild progenitor of cucumber mostly displays late flowering, while the majority of domesticated cucumbers are early flowering. A recent study has narrowed both late-flowering locus (*Lf1.1*) and early-flowering locus (*Ef1.1*) to a 58.8 kb region on chromosome 1 of the 9930 genome, which contains a gene homologous to *Arabidopsis FLOWERING LOCUS T*[11] (*CsFT*; Fig. 5a). We screened our SV dataset and found that one canonical insertion (SV_INS_1G024790, 39.3 kb) and two complex insertions (SV_COMINS_1G013610, 25.3 kb and SV_CO-MINS_1G013600, 44.0 kb) were located 16.5-kb upstream of *CsFT* (Fig. 5b). The 44.0 kb complex insertion was identified in this study and the two other SVs were reported previously[11]. Based on genome sequences of accessions carrying these SVs, we constructed four types of upstream regions (URs) of *CsFT*, long-1, long-2, short-1, and short-2 (Supplementary Fig. 9). The long-1 type of UR contained a 44.7 kb and a 12.1 kb segment, while the long-2 type was identical to the previously defined 'long type' composed of a 39.9 kb and a 12.1 kb sequence[11]. The short-1 UR was present in the 9930 reference genome with only the 12.1 kb segment, while the short-2 type only contained a 37.4 kb fragment (Supplementary Fig. 9). We then proposed an evolutionary route of the *CsFT* locus based on the population-scale SV genotyping enabled by the graph-based pan-genome, days of flowering times, and phylogenetic relationships in the 115-line cucumber population (Fig. 5c). The long-1 type of UR was only present in two Indian wild accessions, which displayed a significantly delayed flowering; therefore, this UR was deemed the ancestral type (Fig. 5c). The long-2 type was proposed to be an outcome of initial domestication, because accessions carrying this UR mainly belonged to Xishuangbanna and Indian groups, which contained landraces and local varieties. It is worth noting that these accessions exhibited high variance in flowering times, suggesting a possible mixture of wild and cultivated cucumbers (Fig. 5c). Accessions harboring short-1 or short-2 types were mainly cultivars showing early flowering. The different geographic distribution of accessions from the two cultivated groups (East-Asian and Eurasian) and structural differences between the two short types of URs suggest a putative independent selection process (Fig. 5c). These results enhance our understanding of the *CsFT* locus evolution during cucumber domestication and global radiation.

**Structural variants associated with cucumber domestication**. We identified 2578 SVs located in regions that have undergone selection during cucumber domestication[21] (domestication-associated SVs, hereafter dSVs) (Supplementary Table 28 and Supplementary Data 5 and 6). To incorporate more SVs that potentially exhibit selection signals, we identified 8651 SVs exhibiting significant frequency changes between wild and cultivated groups (highly divergent SVs, hereafter hdSVs) (Supplementary Table 29 and Supplementary Data 7). Up to 91.7% (4917 insertions, 3003 deletions, and nine inversions) of these hdSVs were not in the dSV set while a proportion (35.2%) of them were in regions showing relatively high ratios (≥3) of nucleotide diversity between wild and cultivated groups. We found that 1611 of these dSVs and hdSVs, of which 131 are within CDS regions and 1480 are within promoter regions, were potentially associated with significantly altered expression of the neighboring genes between wild (Cuc64) and cultivated (9930) cucumbers (Supplementary Table 30 and Supplementary Data 8 and 9).

Root growth of cultivated cucumber is substantially faster than that of its wild progenitor, probably a consequence of intense human selection. To identify SVs that might be associated with this trait, we scanned CDS- and promoter-SVs that potentially impact gene expression in roots during domestication. Interestingly, we found a 76-bp insertion located within the shared promoter of *Csa9930_7G006910* and *Csa9930_7G006920* (SV_INS_7G004090, hereafter pINS) and a 135-bp insertion in the intron of *Csa9930_7G006910* (SV_INS_7G004080, hereafter iINS) (Fig. 6a, b). PCR and PacBio read mapping confirmed these two canonical insertions (Supplementary Figs. 10 and 11). Both genes are homologs of *Arabidopsis AT5G09530* (*PELPK1*) that encodes a positive regulator of root development[46]; thus, they were named *PELPK7.1* (*Csa9930_7G006910*) and *PELPK7.2* (*Csa9930_7G006920*). Haplotype analysis of this two-gene region indicated that three Indian accessions (Cuc64, W4, and W8) formed a distinct cluster and harbored pINS and iINS (Hap1). On contrary, the other cultivated forms did not carry these two insertions (Hap3, the reference type), except for Cuc80 and Cuc37 that harbored pINS (Hap2) (Fig. 6c). In the 115-line population, pINS was found in 19 out of 30 (63.3%) Indian accessions, but only in 11 out of 66 (16.7%) other cultivated cucumbers from East-Asian and Eurasian groups. However, all accessions in the Xishuangbanna group contained pINS, probably due to their monophyletic origin and the narrow genetic basis in this population[21]. iINS was present only in eight of 30 Indian accessions (Fig. 6d). We also observed a clear reduction in terms of nucleotide diversity (π) of this region in cultivated cucumbers compared to wild cucumbers (Supplementary Fig. 12), suggesting a putative selection signal for both SVs. RNA-seq data indicated that expression of *PELPK7.1* and *PELPK7.2* was markedly higher in roots than in other tissues except tendrils in both cultivated (9930) and wild (Cuc64) cucumbers, and both genes were expressed at a significantly lower level in roots of the wild cucumber (Supplementary Fig. 13). Furthermore, expression of *PELPK7.1* and *PELPK7.2* in roots of accessions with Hap1 or Hap2 was significantly lower than that in accessions with the reference genotype (Hap3) (Fig. 6e). Primary root lengths and weights of these accessions were generally in accordance with the expression patterns of *PELPK7.1* and *PELPK7.2*, with lower expression levels linked to shorter primary root lengths and lower root weights (Fig. 6f). Therefore, we propose that cucumber PELPK7.1 and PELPK7.2 are possibly responsible for the differential root development between wild and cultivated cucumbers, and pINS and iINS might be associated with the differential expression of these two genes.

## Discussion

The 11 representative chromosome-scale genome assemblies unraveled a surprising amount of diversity that a single reference cannot represent among cucumber accessions. The construction of a graph-based pan-genome enabled us to accurately genotype SVs at the population scale, facilitating the identification of SVs associated with agronomic traits and helping elucidate possible selection or evolutionary processes. These results present an example of how pan-genomics can enhance our understanding of biologically important questions.

In order to map quantitative trait loci (QTLs) associated with wild cucumber-derived traits such as disease resistance, several attempts have been carried out to create segregating populations by crossing wild and cultivated cucumber lines[20,25,27,47]. However, these studies may have overlooked the large-scale chromosomal rearrangements existing between the parental accessions that result in significant recombination suppression thereby hampering linkage analyses[20,25]. Therefore, our results of presence/absence information of the seven large chromosomal rearrangements among 12 cucumber accessions will benefit

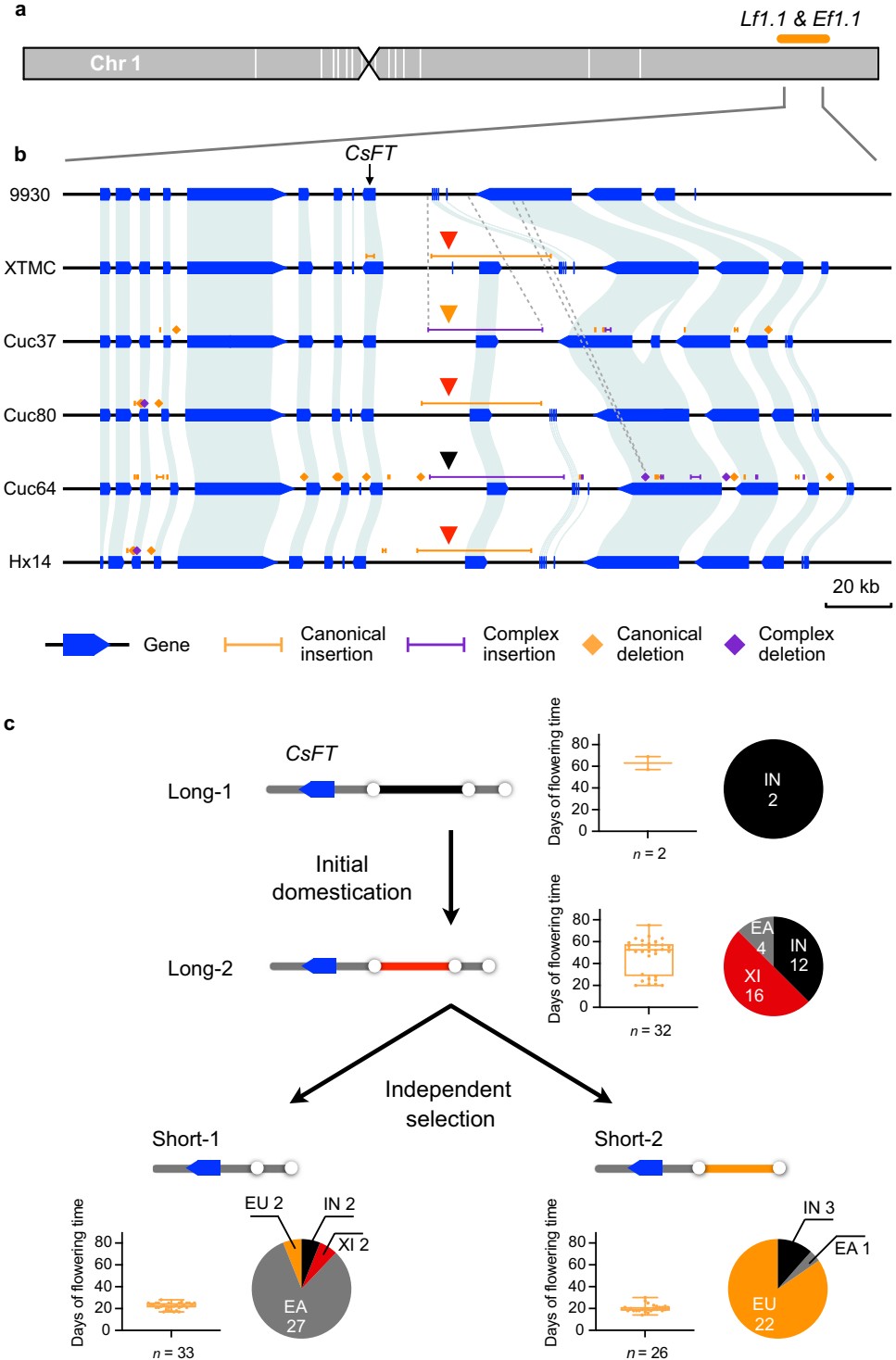

**Fig. 5 Proposed evolutionary trajectory of the *CsFT* locus. a** *Lf1.1* and *Ef1.1* previously mapped to a region on cucumber chromosome 1 of the 9930 reference (the orange bar). **b** Genome region harboring *CsFT* among six cucumber accessions. Breakpoint coordinates in the 9930 reference and the corresponding accession are exhibited by gray dash lines using two SVs as examples. Red, orange, and black triangles pinpoint the 39.3 kb canonical insertion, 25.3 kb complex insertion, and 44.0 kb complex insertion, respectively. **c** Proposed evolutionary model of the *CsFT* locus. Days of flowering time for accessions harboring each of the four UR types (long-1, long-2, short-1, and short-2) are exhibited in boxplots, in which the median and upper and lower quartiles are shown. The whiskers extend to the minimum and maximum of the data. Pie charts denote the number of accessions harboring each UR in four different cucumber groups. Source data are provided as a Source Data file.

practical breeding and genetic studies by guiding the selection of wild accessions used as the parents.

The high-contiguity genome assemblies enabled us to construct large collinear alignment blocks among different accessions and perform intergenomic alignments. We identified more SVs, especially large insertions, compared with a previous study based on short-read mapping[23], offering a more comprehensive understanding of genetic variations in cucumber. Surprisingly,

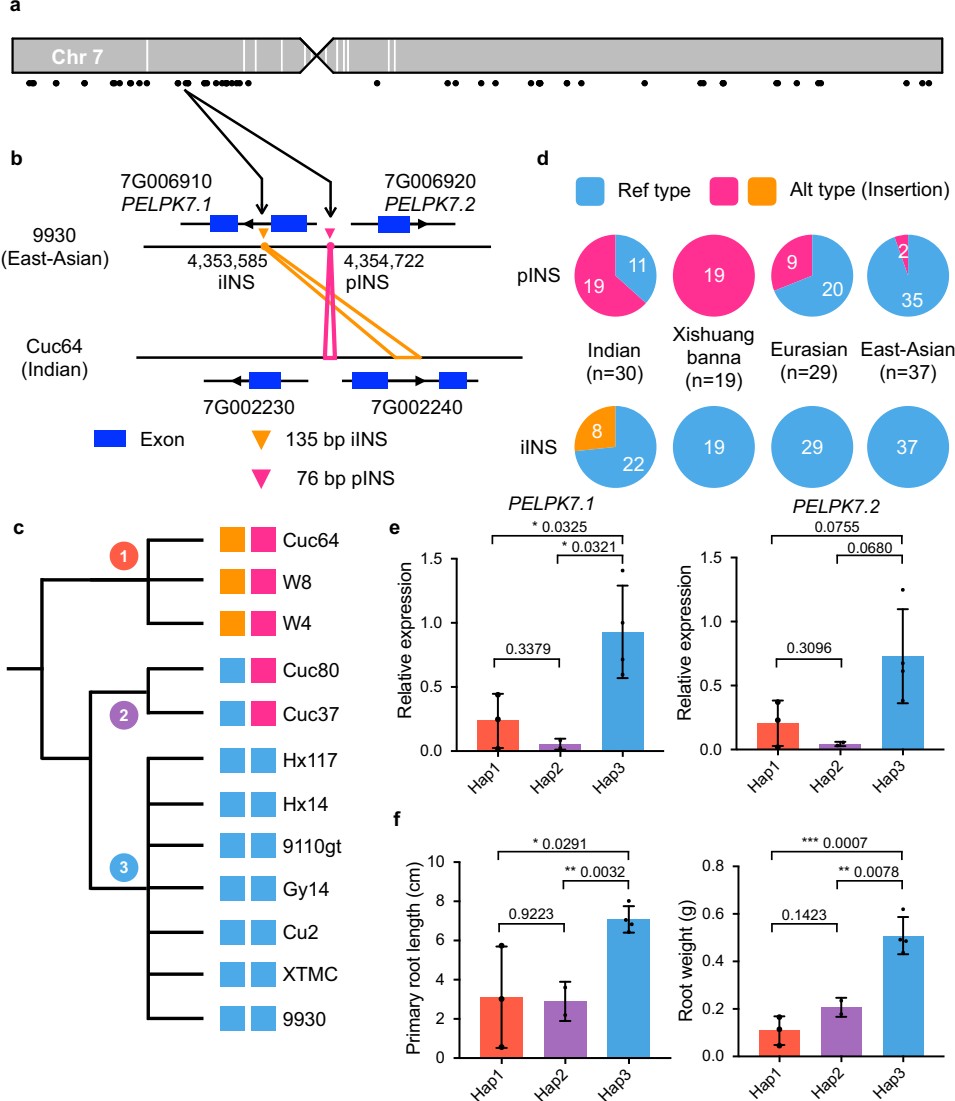

**Fig. 6 Two domestication-related SVs potentially associated with root growth in cucumber. a** Distribution of CDS- and promoter-SVs that might affect gene expression in roots between wild and cultivated cucumbers on chromosome 7. Black points denote SV coordinates. **b** Schematic diagram of two SVs (pINS and iINS) impacting two *PELPK1* genes (*PELPK7.1* and *PELPK7.2*) in cultivated (9930) and wild (Cuc64) cucumbers. **c** Three distinct haplotypes in the 12 cucumber accessions based on the presence or absence of pINS and iINS. **d** Number of accessions carrying reference, pINS and iINS types of alleles in four cucumber groups. **e** Relative expression levels of *PELPK7.1* and *PELPK7.2* in roots of seedlings at 7 days after germination (DAG) in accessions with the three different haplotypes. **f** Primary root length and root weight of seedlings at 7 DAG in accessions with the three different haplotypes. Data are presented as mean ± SD, and numbers of samples (*n*) in haplotype1 (Hap1: Cuc64, W8, and W4), haplotype2 (Hap2: Cuc80 and Cuc37), and haplotype3 (Hap3: Hx117, 9110gt, Cu2, and XTMC) are 3, 2, and 4, respectively. *p*-values are shown in **e** and **f**. **p* < 0.05, ***p* < 0.01, and ****p* < 0.001 in Student's *t*-tests (two-sided). Source data are provided as a Source Data file.

cucumber possesses a markedly fewer number of SVs (3213–21,261 large InDels in each of the 11 accessions, >50 bp in size) compared with other plant species with similar genome sizes, such as Medicago[48] (27,000–110,000 in each of the 15 accessions) and rice[49] (21,752–51,738 in each of the 63 accessions). Cucumber contains seven pairs of chromosomes while other species in the *Cucumis* genus all possess 12, limiting intercross and gene interaction, which has therefore resulted in a narrow genetic background. Furthermore, the domestication bottleneck in cucumber has been reported to be more severe than other crop species[21]. These may contribute to the low diversity within the cucumber species. However, it is also possible that the relatively small sample size in this study could lead to the underestimation of biodiversity. Nonetheless, the methodology applied in this research presents a road map for further studies to

characterize the full spectrum of genetic diversity by assembling genomes from more accessions of this important vegetable crop.

The graph-based pan-genome integrating SV information while preserving the reference genome coordinates offers a powerful platform for cucumber biology and genetic research. Breeders can map their resequencing short reads against the graph-based pan-genome to obtain a precise view of SV landscape in their samples. Further studies should concentrate on these SVs in larger cucumber populations along with high-accuracy phenotypic data, which will allow the breeding community to perform SV-based association studies for trait discovery and improvement. Our study did not incorporate SNPs and small InDels into the graph-based pan-genome, which will be worth further investigating on how the graph structure improves calling or genotyping of these variants. This pan-genome graph

resource will forward plant comparative genomic studies and allow the application of the gene repertoire and SVs for genome-guided breeding and improvement in cucumber.

## Methods

**Sample preparation and sequencing.** A total of 11 accessions comprising two East-Asian lines (XTMC and Cu2), three Eurasian lines (Cuc37, Gy14 and 9110gt), one Xishuangbanna line (Cuc80), and five Indian lines (Cuc64, W4, W8, Hx117, and Hx14) were selected based on the previously constructed phylogeny of a 115-line cucumber core collection[21] (Supplementary Fig. 1). PacBio sequencing was performed for all 11 lines. 10X Genomics and Hi-C sequencing was also performed for three lines (Cuc37, Cuc80, and Cuc64). To evaluate the representativeness of the 11 accessions used in this study plus the 9930 reference, we randomly selected 10,000 SNPs with missing genotype rates <0.1 from a previously genotyped cucumber core collection of 115 lines (3,530,580 SNPs)[21], and repeated this process 20 times. Based on these SNPs, Modified Rogers distance ($MR$), Cavalli-Sforza and Edwards distance ($CE$), and Shannon's diversity index ($SH$) were computed by the 'coreanalyser' program in Core Hunter v2.0 (ref. [50]) using default parameters, and the coverage value ($CV$) of the 12 accessions was calculated using GenoCore[51].

High-quality genomic DNA was extracted from young leaves using a modified CTAB method[52]. Genomic DNA was sheared to a size range of 15–40 kb by a Megaruptor (Diagenode) device, and then used for single-molecule real-time (SMRT) library preparation as recommended by Pacific Biosciences. The constructed SMRT libraries were sequenced on PacBio RSII or Sequel platforms.

A total of 0.3 ng high-molecular-weight DNA was prepared and loaded onto a Chromium Controller chip with 10X Chromium reagents and gel beads, following the recommended protocols. On average, the loaded DNA molecule was ~50 kb in length. There were about 1 million droplets on a Chromium Controller chip. Within each droplet, several DNA molecules were sheared, and the sheared DNA fragments were tagged with the same barcode. All barcoded DNA fragments within these droplets were then sequenced on an Illumina HiSeq X Ten system.

Leaves were fixed with 1% formaldehyde solution, and chromatin was cross-linked and digested using restriction enzyme *Hind* III. The 5' overhangs were filled in with biotinylated nucleotides, and free blunt ends were then ligated. After ligation, crosslinks were reversed and the DNA was purified from protein. Purified DNA was treated to remove biotin that was not internal to ligated fragments. The DNA was then sheared into fragment sizes of ~350 bp. Libraries were prepared using the standard protocol[53] and sequenced on an Illumina HiSeq X Ten platform.

**Genome assembly.** *De novo* assembly of the PacBio reads was performed using CANU[54] (v1.8) with the parameter 'genomeSize=350 m'. The resulting contigs were aligned to bacterial and cucumber plastid genomes using BLASTN v2.2.30+[55], and those with more than 70% of their sequence showing >95% identity with a bacterial or plastid genome were discarded. Pilon v1.22 (ref. [56]) was used to correct the potential base errors in the remaining contigs with parameters: '--fix all --chunksize 20000000 --mindepth 0.4 -K 65 --gapmargin 150000 --vcf --changes --tracks --minmq 10' using previously generated Illumina reads[21]. Corrected contigs were then anchored onto the seven linkage groups of the four genetic maps[25–28] using ALLMAPS[57] with default parameters. Contigs conflicting with the orders of molecular markers from the four genetic maps were manually checked and split when no enough information was available to support the link. The final contigs were then connected into scaffolds using 10X linked-reads by ARCS (v1.0.2)[58] with parameters '-s 98 -c 5 -l 0 -d 0 -r 0.05 -e 30000 -v -m 20-20000'. Finally, Hi-C reads were mapped to the assembled scaffolds using Juicer[59] with default parameters, and 3D-DNA (v180419)[60] was used to cluster and order them into seven chromosome-level super-scaffolds with parameters '-m haploid -i 15000 -r 0'. The four genetic maps mentioned above were further used to orientate the super-scaffolds into the seven pseudo-chromosomes. Whole-genome Hi-C contact heatmaps were plotted using HiCPlotter (v0.6.6)[61] with parameters '-nl 1 -wg 1 -chr whole -fh 0 -r 10000 -hmc 3 -ext pdf'.

PacBio reads were assembled into contigs as mentioned above. The contigs were subsequently anchored onto the seven linkage groups[25–28] of the four genetic maps, and ordered and orientated into the seven pseudo-chromosomes using ALLMAPS[57]. Contigs that could not be anchored onto the seven linkage groups were aligned to the 9930 reference genome using MUMmer[62] (version 4.0.0beta2) with parameters '-maxmatch -c 65 -l 20', and manually placed onto corresponding chromosomes based on the alignment results. BUSCO[29,63] (v5.2.1) was used to evaluate the genome/gene completeness with the embryophyta odb10 database.

**Genome annotation.** RepeatModeler (v1.0.11; http://www.repeatmasker.org/RepeatModeler/) was used to perform *de novo* search for repetitive sequences except LTR-retrotransposons (LTR-RTs) in each genome assembly. Intact LTR-RTs were identified using both LTR_FINDER[64] (v1.0.7) with default parameters, and LTR_harvest (v1.5.10)[65] with following command lines: 'gt suffixerator -db ref -indexname ref -tis -suf -lcp -des -ssp -sds –dna' and 'gt ltrharvest -index ref -similar 90 -vic 10 -seed 20 -seqids yes -minlenltr 100 -maxlenltr 7000 -mintsd 4 -maxtsd 6 -motif TGCA -motifmis 1'. LTR_retriever (v1.6)[66] was then used to filter out false intact LTR-RTs and generate a non-redundant LTR-RT library, using

default parameters. Repeat libraries generated by RepeatModeler and LTR_retriever were merged and RepeatMasker (v1.332; http://www.repeatmasker.org) was applied to carry out repetitive sequence masking.

Based on the '80-80-80' rule[67], TEs (>80 bp) were classified into the same family if they shared ≥80% sequence identity in at least 80% of their coding or internal domain, or within their terminal repeat regions, or in both. The intact LTR-RTs from *Gypsy*, *Copia*, and unknown super-families were then classified into families based on the rule using cd-hit-est (v4.8.1)[68], with the parameters '-c 0.8 -G 0.8 -s 0.8 -aL 0.8 -aS 0.8 -d 0'.

Protein-coding genes were predicted using EVidenceModeler (v1.1.1)[69] by integrating results from ab initio gene predictions, RNA-seq read mapping, and protein homology. RNA-seq data generated from ten tissues of cultivar 9930 (ref. [70]) and seven tissues of wild cucumber PI183967 (ref. [21]) were used for gene prediction. To rescue genes that potentially failed to be predicted in one genome, coding sequences of predicted genes from the other 11 genomes were aligned to the genome assembly using SPALN (v2.3.2)[71] with parameters '-Q7 -O12'. Only alignments with both identity and coverage >95% were kept. Gene models were inferred from these alignments and those that were not present in the initial prediction were integrated.

All predicted proteins were aligned against the UniProt SwissProt (http://www.uniprot.org/) and The *Arabidopsis* Information Resource (TAIR) (https://www.arabidopsis.org) databases. Functions of the best-matched proteins were assigned to the predicted proteins. Functional annotation was also performed using InterProScan[72] (v5.27-66.0) with parameters '-cli -iprlookup -goterms -tsv'. GO terms were assigned according to the InterPro classification.

**Identification of chromosomal rearrangements.** Genome sequences of Cuc37, Cuc80, and Cuc64 were aligned to the 9930 reference using the nucmer program in the MUMmer software[62] (version 4.0.0beta2) with parameters '--maxmatch -c 90 -l 40'. The intergenomic alignment results were manually checked, and exact breakpoint information of the identified large inversions were obtained. For the eight other cucumber accessions (XTMC, Cu2, Gy14, 9110gt, W4, W8, Hx14, and Hx117) without Hi-C-based pseudo-chromosomes, their assembled contigs were aligned to the 9930 genome, and potential breakpoints of the identified large inversions were manually checked based on the alignments.

**Phylogenetic analysis.** All-versus-all alignments of protein sequences from the genomes of 12 accessions were performed using BLASTp[55] with an E-value cut-off of 1e-5. The results were passed to OrthoMCL (v2.0.9)[73] for gene families clustering with the parameter 'percentMatchCutoff = 50'. Protein sequences from gene families with exact one copy from each of the 12 genomes were aligned using MUSCLE (v3.8.31)[74] with the parameter '-maxiters 64', and Gblocks (0.91b)[75] was used to remove poorly aligned regions with default parameters. Alignment results were then fed to IQ-TREE[76] (v1.6.12) to calculate the best-fit amino acid substitution model using the parameter '-m MF', and the consensus maximum likelihood tree of the 12 cucumber accessions was constructed using IQ-TREE with parameters '-m JTT + F + R2 -nt 20 -b 100'.

**Analyses of the cucumber pan-genome.** A total of 299,692 protein-coding genes annotated from the 12 accessions were passed to GET_HOMOLOGUES-EST pipeline (v3.0.9)[77] for gene clustering, with parameters '-M -z -t 0 -C 50 -S 95'. Redundant clusters with ≥ 95% sequence identity and ≥ 50% alignment coverage were eliminated. For each cluster, if a gene from the 9930 reference was present, the gene was selected for further analyses; otherwise, the longest gene was chosen.

RNA-seq reads were mapped to the assembled genomes using HISAT2 (ref. [78]) (v2.1.0) with the '--dta' parameter. Genome-guided transcript assembly and estimation of expression levels of predicted genes were performed using StringTie[79] (v1.3.4) with default parameters.

The frequencies of GO terms in core genes or dispensable genes were compared with those of all non-redundant genes in the 12 cucumber accessions, using Fisher's exact tests in R (v3.6.1)[80], and those with false discovery rate (FDR) < 0.05 were regarded as significantly enriched.

**Identification of genomic variants.** Each genome of the 11 accessions was aligned to the 9930 reference genome using the nucmer program in the MUMmer software[62] (version 4.0.0beta2) with parameters '-maxmatch -c 65 -l 20 -b 500'. The alignment results were filtered using the delta-filter program in MUMmer with parameters '-1 -i 90 -l 200', and only one-to-one alignment blocks were retained. To construct collinear blocks and perform in-block realignment, the axtChain program (v369)[81] with the parameter '-linearGap=medium' and the chainNet program (v369)[81] were subsequently used to build alignment blocks, with a maximum gap length inside one block of 50 kb. For each pair of alignment blocks between the 9930 reference and the corresponding accession, pairwise alignment was performed using the diffseq program (v6.5.7) in the EMBOSS package[82] with parameters '-rformat excel -rusashow3 -wordsize 10'.

Homologous SNPs and small InDels (≤50 bp in size) were extracted from the diffseq outputs. To identify SNPs and small InDels caused by potential base errors in the assemblies, we aligned resequencing reads to the genome of the same accession using BWA (v0.7.17-r1188)[83]. Genomic positions harboring homozygous

variants passing the quality control (base quality > 20, mapping quality > 30, and 2 < read depth < 200) were identified by SAMtools and BCFtools (v1.10.2)[84], and these variants were then removed from our SNP and small InDel dataset. Finally, snpEff (v4.3i)[85] was used to annotate the functional effects of SNPs and small InDels.

Large InDels (≥50 bp in size) were identified from the alignment results of the diffseq program. Variants with precise breakpoints were defined as 'canonical SVs', while those without exact alignment boundaries were classified into the 'complex SV' category. Given that these confusing variants might be caused by the limitation of the diffseq alignment program, their 1 kb left- and right-spanning sequences were extracted, and re-alignments were performed using BLAT (v34x12)[86] with parameters '-noHead -out=psl -t=dna -q=dna'. Variants with precise breakpoints generated by BLAT were added to the 'canonical SV' category. Moreover, SVs containing any unknown sequences were excluded.

Read depth (RD), split-read (SR), and read-pair (RP) analyses were performed using Illumina paired-end sequencing reads to examine the number of deletions that could be supported by short reads. For RD, the Illumina sequencing reads of each accession were mapped to the 9930 reference genome using BWA[83], and the alignments with mapping qualities < 20 were excluded. PCR duplicates were also removed using SAMtools[84], and read depth of each position on the 9930 reference was derived. Average coverages of the deletion region and its 1-kb left- and right-spanning regions were calculated, and the deletion was defined as 'RD supported' if the coverage of the flanking regions was more than threefold of the deletion region and the coverage of the deletion region was <3. For SR and RP, LUMPY (v0.3.1)[87] and DELLY (v0.8.7)[88] were used to identify candidate SVs with default parameters. Potential false SVs were identified and then excluded based on the following criteria: (1) SVs with length < 50 bp; (2) SVs supported by fewer than three RP or SR reads; or (3) SVs with 'LowQual' flags. Deletions that could be detected by any of the approaches above were considered as 'short-read-supported deletions'. For insertions, we transformed an SV from 'insertion' to 'deletion' by swapping the reference genome, since our identified SVs presented precise breakpoints in both genomes being compared. This allowed us to apply the same pipeline deploying on the deletion type of SVs.

The show-coords program in MUMmer[62] was used to extract alignment blocks (parameters '-THrd') from the intergenomic alignment results, and SyRI (v1.0)[89] was used to identify candidate inversions, intra-, and inter-chromosomal translocations. Inversions showing low collinearity in their left- and right-spanning regions, and those that could not be encompassed by a single PacBio contig were manually checked. Translocations carrying 'N' sequences were also removed.

Raw PacBio sequencing reads were mapped to the assembled genome using minimap2 (v2.9-r748-dirty)[90] with parameter '-ax map-pb', and read depth for each genomic position was calculated.

### Graph-based pan-genome construction and population-scale SV genotyping.
Reference sequence of 9930 and 54,107 SVs comprising insertions, deletions, and inversions from the 11 accessions were built into a variant graph using the 'construct' subcommand of vg[17] (v1.23.0) without removing any alternate alleles. The preliminary graph was simplified using 'vg prune', and indexes in XG and GCSA formats were created with 'vg index', in both of which the '-L' parameter was enabled. Illumina paired-end reads of the 115-line cucumber core collection[21] were subsequently mapped against the graph genome and alignments in the GAM format were generated. Alignments with mapping quality <5 or base quality <5 were excluded. Finally, a compressed coverage index was calculated using 'vg pack' and snarls were generated using 'vg snarls', both with default parameters. SV genotyping results from the constructed graph for each of the 115 lines were produced using 'vg call' with the '-v' parameter.

### Genome-wide association studies using SVs.
To estimate the population structure and individual relatedness, we included 54,107 SVs identified in this study and 4,910,802 SNPs identified in the 115-line cucumber population. SVs and SNPs showing missing genotype call frequencies >0.1 or minor allele counts <5 were excluded. A total of 40,720 SVs and 2,833,550 SNPs passing the above-mentioned quality control were used to compute a kinship matrix using the Balding-Nichols method via 'emmax-kin' program in the EMMAX software (v20120205)[91]. We used 11,952 SVs and 205,724 SNPs that were linkage disequilibrium-pruned using PLINK (v1.90b6.22)[92] with parameters '--indep-pairwise 50 5 0.2' to calculate the first ten principal components as co-variants to account for population structure. EMMAX[91] was next applied to perform association tests incorporating genotypes, co-variants, and kinship between SVs and three phenotypes: female flower rate on a primary branch, fruit spine/wart density, and branch number. The genome-wide significance threshold ($3.46 \times 10^{-5}$) was determined by a uniform threshold of $1/n$, where $n$ was the effective number of independent SVs calculated using Genetic type 1 Error Calculator (v0.2)[93].

### Identification of domestication sweeps.
Previously generated resequencing reads from the core collection of 115 cucumber lines[21] were mapped to the 9930 reference genome using BWA[83] with default parameters, and SAMtools[84] was used to perform sorting and duplicate-marking of the alignments. SNPs and small InDels were called using GATK[94] (v3.2-2) and further filtered. SNPs exhibiting

QD < 2.0 or MQ < 40.0 or FS > 60.0 or SOR > 3.0 or MQRankSum < −12.5 or ReadPosRankSum < −8.0 or GQ < 20.0 were removed. The same filtering criteria of SNPs were applied to small InDels except that the thresholds of FS and SOR were set to 200.0 and 10.0, respectively. Ratios of nucleotide diversity ($\pi_w/\pi_c$) and XP-CLR[95] values between wild and cultivated groups were then calculated. Genomic regions showing both the top 5% of $\pi_w/\pi_c$ values and the top 5% of XP-CLR values were considered as domestication sweeps.

### Identification of SVs possibly involved in cucumber domestication.
First, SVs with >50% of sequences overlapping with selective sweeps were defined as domestication-associated SVs (dSVs). Furthermore, the frequencies of each geno-typed SV in the wild and cultivated groups were compared using Fisher's exact tests. Raw $P$ values were subsequently adjusted using the FDR function in R. SVs with FDR < 0.01 were defined as highly divergent SVs (hdSVs).

### Identification of SVs possibly affecting nearby gene expression.
To identify SVs that potentially affected nearby gene expression, we first identified orthologous gene pairs between wild (Cuc64) and cultivated (9930) cucumbers using SynOrth (v1.0)[96] with default parameters and extracted genes close to dSVs or hdSVs (in upstream or genic regions). By calculating fold-changes of expression levels (transcripts per million (TPM)) of these genes between wild and cultivated cucumbers using RNA-seq data from seven tissues (roots, stems, leaves, male flowers, female flowers, fruits, and tendrils), we identified CDS- and promoter-SVs displaying significantly elevated/reduced expression levels ($\log_2$(fold-change) ≥ 1.5 or ≤ −1.5) in at least one tissue between wild and cultivated cucumbers.

### PCR validation of SVs.
Genomic DNA was extracted from fresh leaves, and PCR was performed using 2×Taq PCR Master mix (TIAN GEN). Two SVs within the *PELPEK7.1* and *PELPEK7.2* were analyzed by genotyping PCR using the following primer pairs: GTAGAACTTCTCTCAATGGTTACAC/ATCTTCTTCAATCATA TTTCTCGAAC for iINS and CAATATTGATTAACTTACCTAAT/GAAGACC ATGTAGAGCTTTATA for pINS.

### RNA extraction and quantitative real-time PCR.
Surface-sterilized cucumber seeds were grown on Murashige and Skoog basal medium (MS) with vitamins (Phytote-chlab). Plates were kept in darkness for 2 d before being transferred to a growth chamber with a 16 h light:8 h dark cycle at 25 °C. Total RNA from seedlings at 7 days after germination (DAG) was isolated using the Quick RNA Isolation Kit (Huayueyang). Reverse transcription was performed using GoScript™ Reverse Tran-scription Mix (Promega). Quantitative real-time PCRs (qPCRs) were performed on an CFX96 (Bio-Rad) machine using SYBR® Green Real-time PCR Master Mix (TOYOBO) following the recommendation of manufacturers. Three biological repli-cates were applied in qPCR experiments. Relative gene expression levels were calcu-lated using the Bio-Rad CFX Maestro. Primer pairs used in qPCRs were GGCAATTTCGGCTTTTCCAA/CGTGGATGCGAGGTATCTCT for *PELPK7.1* and ATGTCTACTCATCGCTACTC/GTAATGGTGTCTCCAAAAGA for *PELPK7.2*. The *UBIQUITIN* gene *Csa9930_3G039890* was used as the internal control.

**Reporting summary**. Further information on research design is available in the Nature Research Reporting Summary linked to this article.

## Data availability
Genome assemblies of the 11 cucumber accessions have been deposited in NCBI GenBank under the accession number PRJNA657438. Raw PacBio sequencing data of the 11 cucumber accessions and Hi-C sequencing reads of Cuc37, Cuc80, and Cuc64 have been deposited in the NCBI sequence read archive (SRA) under the accession number SRP278022. Source data are provided with this paper.

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

## Acknowledgements

We thank Prof. Lei Gao (Wuhan Botanical Garden, Chinese Academy of Sciences) and Prof. Jun Yu (Beijing Institute of Genomics, Chinese Academy of Sciences) for critical reading of the manuscript. This work was supported by National Natural Science Foundation of China (32130093 and 31772304 to Z.Z.), China National Key Research and Development Program for Crop Breeding (2016YFD0100307 to Z.Z.), National high-level talent special support Program in China (Z.Z.), the "Taishan Scholar" Foundation of the People's Government of Shandong Province, the Chinese Academy of Agricultural Sciences (ASTIP-CAAS), the Shenzhen municipal (The Peacock Plan KQTD2016113010482651), the US Department of Agriculture National Institute of Food and Agriculture Specialty Crop Research Initiative (2020-51181-32139 to Z.F.), and the "First Class Grassland Science Discipline" program in Shandong Province, China.

## Author contributions

Z.Z. and H.L. conceived and designed the research. S.W., H.L., and Y.X. participated in the plant material preparation. H.L., Z. Yang., Q.Y., S.L., Z. Yao., and X.C. performed the genome assembly and annotation. H.L., Z. Yao., and X.C. performed pan-genome related analyses, intergenomic comparisons, and genetic variant detection. Q.Z. and H.X. identified domestication sweeps. H.L. led in-depth analyses of identified variants associated with phenotypes. S.W. and S.C. prepared the plant samples and performed PCR and qRT-PCR experiments. H.L. and Z.Z. wrote the manuscript. Z.Z., S.H., and Z.F. revised the manuscript. All authors read and approved the manuscript.

## Competing interests

The authors declare no competing interests.
