## [Peer Review File · Nature Communications]

Graph-based pan-genome reveals structural and sequence variations related to agronomic traits and domestication in cucumberEditorial Note: This manuscript has been previously reviewed at another journal that is not operating a transparent peer review scheme. This document only contains reviewer comments and rebuttal letters for versions considered at Nature Communications.

Reviewers' Comments:

Reviewer #1:

Remarks to the Author:

The revised manuscript by Hongbo Li and colleagues reads much, much better than the previous version submitted to Nature Plants which I read back in March 2021. The sentences flow better, and the data is presented in a more coherent, succinct and easy to understand manner. Upon reading the new version of the manuscript, I also felt that some of the grinding statements have now been scaled back, or supported by new analysis, e.g. the extrapolation of structural variants to the whole panel of 115 accessions and showing that some of these variants correlate with phenotype, e.g. as mediated by CsTu.

Notwithstanding, I still have one major concern. I previously pointed out that 12 genomes may not be enough to present the pan-genome of the species and that this should ideally be tested in a "diminishing-return" experiment in which "genetically diverse accessions" from a "previously published available high-throughput genotyping of a large panel" would show that as more genotypes are added then "progressively fewer and fewer novel genetic variants are found." To address this point, the authors have shown that the 12 lines are genetically diverse based on their distribution in the phylogenetic tree (Supplementary Figure 1) and that in a diminishing return graph (Supplementary Figure 5) the horizontal asymptote levels out with these 12 accessions. However, the real analysis should include, as I attempted to suggest a much larger panel, for example, the entire set of 112 already genotyped accessions presented in Figure 1. If the analysis of this 'complete' population shows that the 12 accessions capture e.g. 95% of the genotypes, then it would be fair to claim that the whole pan-genome has been captured. The authors could also, use Core Hunter (Thachuck et al. 2009, Core Hunter: an algorithm for sampling genetic resources based on multiple genetic measures. BMC Genomics 10:243) to analyze what percentage of the total diversity present in the 115 accessions is captured by the 12 accessions selected for high quality genome assembly. In the absence of these analyses the authors cannot claim that they have sequenced and assembled the cucumber pan-genome.

The authors have the data to perform this analysis and it should not take long. Until then I cannot fully support the 'pangenome' statements.

Minor points:

Lines 152-153: Can you really assign priority in terms of the stepwise evolution of the SVs?

Line 220: the GWAS based on SVs identified positive signals, however, these signals could easily be from other genetic variants that are in LD (i.e. 'hitchhiking'). Functional analysis would be required to determine is the SV is causative.

Typos etc:

Line 53: "capturing full" should be "capturing the full"

Line 79: "and additional" should be "and an additional"

Line 93: "represent nearly" should be "represent the nearly"

Line 128: fix grammar
Line 221: "closed" should be "close"
Line 367: "example how" should be "example of how"
Line 393: "Furthermore, domestication" should be "Furthermore, the domestication"
Line 534: "when no enough" should be "when not enough"
Line 619-620: "by following" should be "by the following"
Line 669: "inversing shown" should be "inversions showing"
Line 713: "exclude" should be "excluded"
Line 740: "preformed" – I presume this should be "performed".

Reviewer #2:

Remarks to the Author:

They authors have addressed some of my comments. However, I'm still concerned about the following points:

1. In the section titled "Functional impact of structural variations", line 264-266, the explanation of functional importance and selection of the newly identified SV is weak. It's very hard to understand how "some Indian cultivated lines contained the 51-bp deletion in CsTu" implicated "its functional importance and thus being selected by local breeders". Additionally the conclusion of this paragraph "These results suggest that SVs are important in regulation of fruit spine and wart development and that they have been under differential selection worldwide in response to consumer preferences" is not supported by the evidence presented. There is no direct evidence supporting the roles of SVs in regulation fruit spine and wart development in this study. The claimed "selection" was just evidenced through the low frequency of these SV in some subgroups of cultivated accessions, which is not convincing. If these SV were functionally important and the traits were indeed under selection, it is not clear why the frequency of these SV would be low in the subgroup of cultivated accessions (Figure 4). Could there be other genetic variation involved in genetic control of spines and warts? Were these SV identified under selection also in the genome-wide analysis of SV association with domestication?

2. In lines 151-159, in my opinion the authors extrapolated too much with the explanation of a stepwise evolution of inversions. These seven inversions were not genetically linked, so independent segregation of them in the wild population means they can appear in combinations as the authors observed in the three wild accessions, one carrying 7 inversions (3 in CHR4, 3 in Chr5 and 1 in Chr7), one carrying 3 inversions (3 in Chr5) and one carrying none. This is just a natural variation of these loci. If the authors genotype more wild accessions, it is very likely that there would also be wild accessions carrying 1 inversion, 4 inversions, 6 inversions etc.

3. line 306-321, the overlap between the identified "dSVs" and "pdSVs" is very low. The authors need to explain why most "dSVs" were not identified in "pdSVs". If both methods are correct, most "dSVs" should be identified as part of "pdSVs".

4. The writing of this manuscript still requires work. There are many inaccurate/incorrect sentences that need to be carefully revised throughout the manuscript.

Such as, line 56-57, change "SVs may play critical roles in plant gene and QTL mapping" to "SVs play critical roles in genome evolution and genetic control of agronomical traits in plants". check the reference "Exploring and exploiting pan-genomics for crop improvement"

Line 58-61, "Recent pan-genome studies in human and plant species have uncovered the species-wide biodiversity in terms of either nucleotides or protein coding genes that a mere reference genome cannot capture, several of which have also characterized SVs by inter-genomic comparison". This sentence needs to be rephrased

Lines 126-128, "Annotation of repeat elements in the 12 genome assemblies (11 accessions and the 9930 reference) resulted in TE contents ranging from 32.5-38.5%, varying in sizes of genome assemblies". Why "annotation of repeat elements" varies "in sizes of genome assemblies" ?

line 113-117, "the high consistency between assembled sequences and Hi-C data indicate that our genome assemblies are of high accuracy" doesn't belong to this sentence. This sentence was describing the assembling of other 8 genomes without Hi-C data. The description of using Hi-C data to assemble three genomes was in line xxx-xxx. Was the "high consistency between assembled sequences and Hi-C data" in the three genomes expected as a result of that? If yes, this doesn't say anything about quality of genome assemblies.

line 122-124, BUSCO is measure of completeness. High BUSCO value does not fully suggest the "qualities" of these gene models are sufficient for downstream analyses.

line 246-247, "Phylogenetic analysis using SNPs within these six genes suggests an Indian ancestor of these loci with Cuc64 and W8 showing small and low-density fruit spines and warts". It is hard to understand how "the phylogenetic analysis" suggests "an Indian ancestor shows small and low-density fruit spines and warts."

line 269-270, "accessions carrying the 10-bp upstream allele displayed significantly elevated fruit spine density, which is indeed preferred in several Asian countries; thereby being retained in some cultivars." This sentence is not correct

"accessions carrying the 10-bp upstream allele" couldn't be "retained in some cultivars". I think what the author wanted to say is the "10-bp upstream allele" was "retained in some cultivars". But, is "retain" the right word? This "10-bp upstream allele" didn't show up in the wild species. So it looks more like a new mutation in the cultivated, rather than something retained from wild.

Line 279-281, it looks the three SV with different coordinates were located at the same spot. Is it right?

Reviewer #3:

Remarks to the Author:

The authors have adequately addressed my previous questions. Several other minor suggestions:

Line 219, the authors performed GWAS with the genotyped SVs in the 115-line cucumber population. I wonder why they did not include SNPs. The authors mentioned in the methods that they used LD-pruned SVs to calculate the first ten principal components as covariates, but which variants were used for kinship matrix is unclear. SVs only capture parts of genomic variation among the population, using SNPs plus SVs for calculating population structure and kinship matrix would represent a better population variation.

Line 300, the authors may need to describe short-1 and short-2 types of URs first, as they described for long-1 and long-2 types of UR in Line 285-287.

Reviewer #4:

Remarks to the Author:

This work constructed a graph-based pan-genome for cucumber by de novo sequencing of 11 wild and cultivated cucumbers and identified potential SVs for agronomic traits and domestication. The fact that this study lacks of novelty as raised by other reviewers is objective, nevertheless, it is also obviously of great importance for cucumber breeders and research community. Following the review process, it is seen that the manuscript has been carefully revised according to the previous reviewers, most of the reviewers' concerns have been addressed in the revised manuscript. Therefore, I don't have many concerns, but I do have some suggestions as listed below.

1. the previous reviewer #2 have the concerns about using resequencing data to evaluate the SVs identified in the pan-genome. The authors haven't given enough explanation in the response letter about why they acknowledged 'the limitation of using resequencing data to identify SVs in the introduction, but later used this method in the analysis.

2. line 317: I don't think the identification of SVs within CDS and putative promoter regions that associated with the change of expression of the closest gene can be evidence of concluding functions during domestication.

3. line 377-382, as suggested by reviewer#3, the authors have added the comparison of their results of structural variations and pan-genome with other plant species, however, the description is bland and unconstructive.

4. line 387 the number of SVs in cucumber (53912) is not fewer than Medicago (27000-), at least not all the Medicago accessions, the statement must be concise and accurate.

5. Supplementary Fig. 3, would you explain why the interaction signal is significantly low in somewhere chr2 in all the three accessions?

Reviewer #5:

Remarks to the Author:

In this study, Li et al constructed a "pan-genome" of 12 cucumber lines, representing both wild and domesticated accessions. The authors put special emphasis on the identification of structural variations – both from the evolutionary perspective and explored the phenotypic consequences of few of these. The manuscript is written generally clearly and presents a large body of work – including the combination of several sequencing technologies to construct chromosome-scale assemblies of these 12 lines. I believe the study and resource will be of interest to researchers outside the immediate field and should span those interested in genomics and plant domestication. Still, I do not agree with some of the interpretations, some are quite central – specifically all those related to the evolutionary pathways proposed – and I suggest the authors carefully reexamine these sections – please see my detailed comments below (General comments #2 & #3, specific comments .

General comments:

1. The term "pan-genome" has been used rather loosely in the literature, from the analysis of very few accessions to hundreds, and as such this study is certainly on the small side. Based on simulation of pan-genome size, the authors argue the number of pan-genes reached a plateau when as low as nine accessions are used. Looking at Figure S5, it doesn't seem as if a plateau was reached. However, this also depends on how a plateau is defined, which was not defined by the authors. Arguing that the plateau was reached based on the plot is problematic since this depends on the resolution of the axes.

As an alternative, it would be informative to specify the number of new genes that were added as more samples were included.

2. Karyotype evolution of cucumber (lines 148-159): I do not agree with this interpretation. Based on the phylogeny of figure 1, we can deduce that the inversion on chromosomes 4 & 7 occurred after divergence between W8 and Cuc64, and along the lineage leading to Cuc64. However, the phylogenetic location of the inversion on chromosome 5 is not clear, since the ancestral lineage could have possessed the karyotype of W8 or that of 9930 – in both cases, a single transition in either of the two ancestral lineages could lead to the observed data. Thus, the interpretation presented in lines 151-159 is one of two equally-parsimonious scenarios.

3. The analyses presented in figure 4b (and in lines 246-248) should be performed on the same phylogeny that is presented in figure 1 – why basing the analysis on a phylogeny derived from the small number of SNPs contained within these 6 genes? The same is true for the tree (and the corresponding analysis) presented in figure 6c.

4. It is unclear whether the analysis can distinguish translocations from large insertions. Can the authors comment on that?

Specific comments:

5. Introduction. I don't think most readers are aware of the term "graph-based pan-genome". A more detailed explanation should help here.

6. Lines 69-71. Rephrase this sentence.

7. Line 77. How does the quality of the genome of line 9930 compares to the genomes produced here?

8. Line 79 . For the 11 representative accessions – should mention here how many wilds and how many cultivated lines.

9. First page of the results - regarding the assembly strategy. The authors integrated several sequencing techniques to assemble these 11 lines, but the details are given as bits and pieces. Similarly, the coverage and genome quality statistics are separated into the different platforms (lines 99-100, 107-108, 110, 115) which is quite confusing. A paragraph that presents the overall assembly strategy could help the readers follow what and why was done.

10. Line 117 - could be helpful to report the % complete BUSCOs on genome assemblies, not just genome annotations. Also – what was the BUSCO score of the reference 9930 genome?

11. Line 121 – what is the CDS length of these genes?

12. Lines 125-136. Aside from this section, no analyses were performed with regards to transposable elements. With develop this analysis more thoroughly or else - I think this paragraph could be substantially shortened and appended to the previous paragraph.

13. Lines 144-145. This sentence is unclear. What does it mean "discrete chromatin interactions around their breakpoints"?

14. Line 180. The KaKs analysis does not suggests that dispensable genes have undergone greater positive selection, but that dispensable genes have undergone less stringent purifying selection.

15. Lines 196-197. The definition of complex InDels is confusing. It would be helpful to better describe these and what type of actual mutation they may represent.

16. Line 191 - how robust are the results to the choice of 'pivot' genome? What if you chose another one rather than the reference? What would be the effect on the catalog of SVs?

17. Line 214 - can the catalog of SVs and/or the SV graph be used to detect gene PAV? It could be interesting to compare this to the results obtained in the previous section by annotation and clustering of genes.

18. Line 219. It is unclear against what phenotype the GWAS was performed. I see that this information is presented in the Supp Materials, but it should also be noted in the main text.

19. Lines 293-295. It is not clear why lines CG0001 and CG0002 are considered as the ancestral lines – the ancestor types could have been W4 just as well.

20. Line 308. Change to "located in regions that have undergone domestication sweeps "

21. Line 317. "131 ARE within CDS regions and 1,480 ARE within..."

22. Line 366: "evolutionary processes"

23. Line 371: "these studies may HAVE overlooked"

24. Lines 379-381. The proportion of core genes greatly depends on the exact definition of core genes used in each study and on the number of accession used in each pan-genome. Thus, this comparison should be re-made while using the same definition for all pan-genomes. For example, the percentage reported for brachipodium of 54% includes also genes that are missing from one accession.

25. Line 395 - it is quite possible that the low genomic diversity observed in cucumbers is the result of the methods applied in this study and the low number of samples - this (along with other limitations of the method) should be discussed here.

26. Figure 3 - I find it confusing that panels (d) and € are displayed as heat maps whereas (f) and (g) are displayed as line plots, while they describe similar things.

27. Figure 3 - I found panels (h) and (i) quite confusing. Should better explain what is shown.

28. Figure 4c - the IGV screenshot will probably be difficult to understand for readers who don't regularly use this software. I don't think showing this plot is necessary, but for sure another way to display this is needed.

Reviewer notes:**Reviewer #1: (Remarks to the Author):**

The revised manuscript by Hongbo Li and colleagues reads much, much better than the previous version submitted to Nature Plants which I read back in March 2021. The sentences flow better, and the data is presented in a more coherent, succinct and easy to understand manner. Upon reading the new version of the manuscript, I also felt that some of the grinding statements have now been scaled back, or supported by new analysis, e.g. the extrapolation of structural variants to the whole panel of 115 accessions and showing that some of these variants correlate with phenotype, e.g. as mediated by CsTu.

Notwithstanding, I still have one major concern. I previously pointed out that 12 genomes may not be enough to present the pan-genome of the species and that this should ideally be tested in a “diminishing-return” experiment in which “genetically diverse accessions” from a “previously published available high-throughput genotyping of a large panel” would show that as more genotypes are added then “progressively fewer and fewer novel genetic variants are found.” To address this point, the authors have shown that the 12 lines are genetically diverse based on their distribution in the phylogenetic tree (Supplementary Figure 1) and that in a diminishing return graph (Supplementary Figure 5) the horizontal asymptote levels out with these 12 accessions. However, the real analysis should include, as I attempted to suggest a much larger panel, for example, the entire set of 112 already genotyped accessions presented in Figure 1. If the analysis of this ‘complete’ population shows that the 12 accessions capture e.g. 95% of the genotypes, then it would be fair to claim that the whole pan-genome has been captured. The authors could also, use Core Hunter (Thachuck et al. 2009, Core Hunter: an algorithm for sampling genetic resources based on multiple genetic measures. BMC Genomics 10:243) to analyze what percentage of the total diversity present in the 115 accessions is captured by the 12 accessions selected for high quality genome assembly. In the absence of these analyses the authors cannot claim that they have sequenced and assembled the cucumber pan-genome.

The authors have the data to perform this analysis and it should not take long. Until then I cannot fully support the ‘pangenome’ statements.

Response: We thank the reviewer for the comments and suggestions! We have used Core Hunter 3 (ref ¹) to examine whether the 12 accessions used for high-quality genome assemblies could capture a substantial amount of genetic diversity in the cucumber population of 115 accessions. We randomly selected 10,000 SNPs with missing genotype call rates < 0.1 from the previously genotyped cucumber population, and compared genetic distance measures (Modified Rogers distance, *MR* and Cavalli-Sforza and Edwards distance, *CE*) and community diversity index (Shannon's diversity index, *H*) between the 12 accessions sampled in this study and the 115 cucumber lines. The results indicated that the 12 lines showed a larger *MR* and *CE* (0.495 and 0.498) compared with the 115 accessions (0.375 and 0.380) while exhibited a slightly reduced *H* (9.438 versus 9.508; 0.74% decrease), indicating that these 12 accessions are genetically diverse and capture the vast majority of diversity within the 115-line cucumber population. Furthermore, as suggested by Reviewer #5, we computed the number of new gene clusters that were added when additional accessions were included, and observed that it declined rapidly with only 98 new clusters detected when the number of accessions reached 12 (**Response Fig. 1**). This result has been integrated into the main text (**Line 162-165**).

Response Fig. 1. Number of gene clusters that are newly added when including more accessions.

Minor points:

Lines 152-153: Can you really assign priority in terms of the stepwise evolution of the SVs?

Response: According to the comments from Reviewers 1, 2 & 5, we rephrased this section to interpret karyotype evolution of cucumber more accurately (**Line 143-147**): “Based on the phylogeny of the 12 accessions (**Fig. 1d**), we can deduce that the inversions on chromosomes 4 and 7 could have occurred after the divergence between W8 and Cuc64. Considering that all cultivated and some wild accessions do not possess any inversions on chromosome 5, we propose that these inversions could have occurred during the evolution of wild cucumbers. The breakpoints and presence/absence information of these seven chromosomal rearrangements among the 12 accessions provide novel insights into karyotype evolution in the *Cucumis* genus. Since large segmental inversions can lead to recombination suppression in these regions ², our inversion map of the 12 cucumber accessions provides a guide for properly selecting parental lines to construct segregating populations between wild and cultivated cucumbers.”

Line 220: the GWAS based on SVs identified positive signals, however, these signals could easily be from other genetic variants that are in LD (i.e. ‘hitchhiking’). Functional analysis would be required to determine if the SV is causative.

Response: We totally agree with the reviewer that SVs with association signals may not be causative variants. This is true for all GWAS studies with any types of variants (SVs and/or SNPs). In this section, we attempted to prove that our graph-based genome is a useful platform to genotype SVs in large populations to perform SV-based GWAS. We therefore listed results of three agronomic traits using SV-GWAS by genotyping the previously reported 115-line cucumber population. Among them, we identified peaks containing SVs close to *m* and *F*, two known functional genes that were reported to be involved in cucumber sex determination. These SVs proximal to these genes might not be causative variants. For fruit spine/wart density, we directly identified the

causative SV that was experimentally validated in a previous study³. Regarding the branch number trait, we detected a 2,593-bp insertion upstream of an *Arabidopsis* homolog (*BYPASSI*) that controls plant architecture, providing useful information for future functional characterization of candidate genes. These results demonstrated how the graph-based pan-genome can facilitate the SV-based GWAS of important agronomical traits. While we agree with the reviewer that functional analysis to validate the candidate causative variants would provide critical insights into genetic mechanisms underlying the regulation of the traits, we believe this is beyond the scope of this study.

Typos etc:

Line 53: “capturing full” should be “capturing the full”

Line 79: “and additional” should be “and an additional”

Line 93: “represent nearly” should be “represent the nearly”

Line 128: fix grammar

Line 221: “closed” should be “close”

Line 367: “example how” should be “example of how”

Line 393: “Furthermore, domestication” should be “Furthermore, the domestication”

Line 534: “when no enough” should be “when not enough”

Line 619-620: “by following” should be “by the following”

Line 669: “inversing shown” should be “inversions showing”

Line 713: “exclude” should be “excluded”

Line 740: “preformed” – I presume this should be “performed”.

Response: We have corrected these errors based on the reviewer’s comments. Thanks.

Reviewer #2: (Remarks to the Author):

The authors have addressed some of my comments. However, I'm still concerned about the following points:

1. In the section titled “Functional impact of structural variations”, line 264-266, the explanation of functional importance and selection of the newly identified SV is weak. It's very hard to understand how “some Indian cultivated lines contained the 51-bp deletion in *CsTu*” implicated “its functional importance and thus being selected by local breeders”. Additionally, the conclusion of this paragraph “These results suggest that SVs are important in regulation of fruit spine and wart development and that they have been under differential selection worldwide in response to consumer preferences” is not supported by the evidence presented. There is no direct evidence supporting the roles of SVs in regulation fruit spine and wart development in this study. The claimed “selection” was just evidenced through the low frequency of these SV in some subgroups of cultivated accessions, which is not convincing. If these SV were functionally important and the traits were indeed under selection, it is not clear why the frequency of these SV would be low in the subgroup of cultivated accessions (Figure 4). Could there be other genetic variation involved in genetic control of spines and warts? Were these SV identified under selection also in the genome-wide analysis of SV association with domestication?

Response: We thank the reviewer for pointing these out. To avoid confusion, we have removed the sentence “some Indian cultivated lines contained the 51-bp deletion in *CsTu*, implicating its functional importance and thus being selected by local breeders”. We have also re-written the conclusion of this paragraph to be more accurate: “Our analyses identified a new SV that affects the CDS of *CsTu*, which is worthy of further investigation to examine whether this SV impacts the fruit wart phenotype. These results also reveal the selection landscape of SVs in functionally important genes involved in the regulatory network of cucumber fruit spine and wart development.”
(Line 265-269)

Regarding the “selection” of SVs, traits relevant to cucumber fruit spines and warts exhibit a differentiated consumer preference: fruits of US and European cucumber varieties are mostly non-warty and spine-free, while both warty and non-warty cucumbers are favored in many Asian countries. A proportion of East-Asian cultivars

are spine-rich with large fruit warts. Previous studies have also indicated that the warty phenotype is dominant over the non-warty fruit trait⁴. Loss of function of *CsTu* due to the 4,895-bp deletion has previously been reported to lead to the non-warty phenotype⁴, and presence of the 10-bp upstream segments of *CsGL3* is responsible for the high density of fruit spines³. Considering that our 115-line cucumber population includes numerous natural germplasm that did not undergo artificial selection, it is reasonable to observe relatively low presence frequencies of these SVs.

There could be other genes underlying fruit wart and spine development in cucumber; nevertheless, here we only focused on SVs present in known functional genes. We found that the two SVs located within *CsTu* were potentially under selection during cucumber domestication (these SVs do reside in a genomic region that underwent selection during cucumber domestication), while no signal of domestication was detected regarding the SV upstream of *CsGL3*.

2. In lines 151-159, in my opinion the authors extrapolated too much with the explanation of a stepwise evolution of inversions. These seven inversions were not genetically linked, so independent segregation of them in the wild population means they can appear in combinations as the authors observed in the three wild accessions, one carrying 7 inversions (3 in CHR4, 3 in Chr5 and 1 in Chr7), one carrying 3 inversions (3 in Chr5) and one carrying none. This is just a natural variation of these loci. If the authors genotype more wild accessions, it is very likely that there would also be wild accessions carrying 1 inversion, 4 inversions, 6 inversions etc.

Response: Thanks for the comments! We have rephrased this section to interpret karyotype evolution of cucumber more accurately (**Line 143-147**). “Based on the phylogeny of the 12 accessions (**Fig. 1d**), we can deduce that the inversions on chromosomes 4 and 7 could have occurred after the divergence between W8 and Cuc64. Considering that all cultivated and some wild accessions do not possess any inversions on chromosome 5, we propose that these inversions could have occurred during the evolution of wild cucumbers. The breakpoints and presence/absence information of these seven chromosomal rearrangements among the 12 accessions provide novel insights into karyotype evolution in the *Cucumis* genus. Since large segmental inversions can lead to recombination suppression in these regions², our inversion map

of the 12 cucumber accessions provides a guide for properly selecting parental lines to construct segregating populations between wild and cultivated cucumbers.”

3. line 306-321, the overlap between the identified “dSVs” and “pdSVs” is very low. The authors need to explain why most “dSVs” were not identified in “pdSVs”. If both methods are correct, most “dSVs” should be identified as part of “pdSVs”.

Response: The dSVs were defined as SVs that were localized in regions undergoing selection during cucumber domestication, while the pdSVs were SVs that displayed significant changes of presence frequencies between wild ($n = 30$) and cultivated ($n = 85$) cucumber groups. The method used to identify pdSVs was referenced from Zhang et al.⁵. Theoretically, the domestication sweeps were identified by computing the change of nucleotide diversity based on SNPs between wild and cultivated cucumber groups, while pdSVs were identified as those with highly divergent frequencies between wild and cultivated cucumber groups. Among dSVs within domestication sweeps, many show similar presence frequencies between wild and cultivated groups. Thus, these SVs would not be identified as pdSVs. To avoid confusion, we have changed “pdSVs” to “highly divergent SVs between wild and cultivated groups (hdSVs)” (**Line 310**).

4. The writing of this manuscript still requires work. There are many inaccurate/incorrect sentences that need to be carefully revised throughout the manuscript.

Such as, line 56-57, change “SVs may play critical roles in plant gene and QTL mapping” to “SVs play critical roles in genome evolution and genetic control of agronomical traits in plants”. check the reference “Exploring and exploiting pan-genomics for crop improvement”

Response: We have revised this sentence according to the reviewer’s comment (**Line 55-56**) and added a citation ([https://www.cell.com/molecular-plant/fulltext/S1674-2052\(18\)30383-6](https://www.cell.com/molecular-plant/fulltext/S1674-2052(18)30383-6)).

Line 58-61, “Recent pan-genome studies in human and plant species have uncovered the species-wide biodiversity in terms of either nucleotides or protein coding genes that

a mere reference genome cannot capture, several of which have also characterized SVs by inter-genomic comparison”. This sentence needs to be rephrased

Response: We have changed this sentence to “Recent pan-genome studies in human and plant species have uncovered the species-wide biodiversity with a special emphasis on the characterization of SVs” (**Line 57-59**).

Lines 126-128, “Annotation of repeat elements in the 12 genome assemblies (11 accessions and the 9930 reference) resulted in TE contents ranging from 32.5-38.5%, varying in sizes of genome assemblies”. Why “annotation of repeat elements” varies “in sizes of genome assemblies” ?

Response: We initially proposed that the difference of TE contents among the 12 genomes could be the outcome of different assembly sizes (thus this sentence had grammatical mistakes). Since this description is not important for this section, and as suggested by Reviewer #5, we have removed “varying in sizes of genome assemblies” in the revised manuscript (**Line 116-121**) and integrated this paragraph into the previous one.

line 113-117, “the high consistency between assembled sequences and Hi-C data indicate that our genome assemblies are of high accuracy” doesn’t belong to this sentence. This sentence was describing the assembling of other 8 genomes without Hi-C data. The description of using Hi-C data to assemble three genomes was in line xxx-xxx. Was the “high consistency between assembled sequences and Hi-C data” in the three genomes expected as a result of that? If yes, this doesn’t say anything about quality of genome assemblies.

Response: To make it more clear, we have rephrased this sentence to “The high consistency between assembled sequences of Cuc37, Cuc80 and Cuc64 and corresponding Hi-C data indicate that these genome assemblies are of high accuracy”. Furthermore, we have substantially re-written this section, as suggested by Reviewer #5, and added a description of BUSCO results of the 11 assembled genomes after “These assemblies had total lengths ranging from 232.5 Mb to 251.1 Mb”. Please see the revised manuscript from **Line 113** to **Line 115**. The BUSCO scores have been updated based on the new version (v5.2.1) of the software and the newly released database `embryophyta_odb10`.

line 122-124, BUSCO is measure of completeness. High BUSCO value does not fully suggest the “qualities” of these gene models are sufficient for downstream analyses.

Response: We have revised this sentence to “The mean BUSCO ⁶ score of the predicted genes was estimated to be around 96.0%, suggesting that the predicted gene models in these genomes are sufficient for downstream analyses” (**Line 125-126**). The BUSCO scores have been updated based on the new version (v5.2.1) of the software and the newly released database `embryophyta_odb10`.

line 246-247, “Phylogenetic analysis using SNPs within these six genes suggests an Indian ancestor of these loci with Cuc64 and W8 showing small and low-density fruit spines and warts”. It is hard to understand how “the phylogenetic analysis” suggests “an Indian ancestor shows small and low-density fruit spines and warts.”

Response: We have changed this sentence to “Phylogenetic analysis using SNPs within these six genes suggests that the two Indian accessions (Cuc64 and W8), whose fruits show small and low-density spines and warts, possess the ancestral state of these loci” (**Line 241-243**).

line 269-270, “accessions carrying the 10-bp upstream allele displayed significantly elevated fruit spine density, which is indeed preferred in several Asian countries; thereby being retained in some cultivars.” This sentence is not correct “accessions carrying the 10-bp upstream allele” couldn’t be “retained in some cultivars”. I think what the author wanted to say is the “10-bp upstream allele” was “retained in some cultivars”. But, is “retain” the right word? This “10-bp upstream allele” didn’t show up in the wild species. So it looks more like a new mutation in the cultivated, rather than something retained from wild.

Response: We have revised this sentence to “accessions carrying the 10-bp upstream allele displayed significantly elevated fruit spine density, which is indeed favored in several Asian countries. This could be the reason that this allele has been selected in some East-Asian cultivars” (**Line 262-265**).

Line 279-281, it looks the three SV with different coordinates were located at the same spot. Is it right?

Response: Yes, these three SVs were fairly close, with the genomic coordinates of 29,468,355, 29,469,062 and 29,469,066 bp, respectively, on chromosome 1 of the 9930 reference genome.

Reviewer #3: (Remarks to the Author):

The authors have adequately addressed my previous questions. Several other minor suggestions:

Line 219, the authors performed GWAS with the genotyped SVs in the 115-line cucumber population. I wonder why they did not include SNPs. The authors mentioned in the methods that they used LD-pruned SVs to calculate the first ten principal components as covariates, but which variants were used for kinship matrix is unclear. SVs only capture parts of genomic variation among the population, using SNPs plus SVs for calculating population structure and kinship matrix would represent a better population variation.

Response: Thanks for the suggestion. In this section, we attempted to prove that our graph-based genome is a useful platform to genotype SVs in large populations, which can further facilitate SV-based GWAS. Therefore, we did not include SNPs in the GWAS. In the original manuscript, we computed the kinship matrix using SVs passing quality control with missing genotype call frequencies ≤ 0.1 and minor allele counts ≥ 5 . As recommended by the reviewer, we have re-performed principal component analysis and computed kinship matrix using both SNPs and SVs passing the abovementioned criteria. Based on these results, we re-performed GWAS and found that the peak SVs listed in the main text remained significant with slightly changed p values (as expected). We have updated the results in the main text (**Line 219**) and **Supplementary Fig. 7** and have also added these technical details in the “**Genome-wide association studies using SVs**” of the **Methods** section (**Line 696-711**).

Line 300, the authors may need to describe short-1 and short-2 types of URs first, as they described for long-1 and long-2 types of UR in Line 285-287.

Response: We have added a sentence to define the short-1 and short-2 types of URs in the revised manuscript (**Line 283-285**).

Reviewer #4 (Remarks to the Author):

This work constructed a graph-based pan-genome for cucumber by de novo sequencing of 11 wild and cultivated cucumbers and identified potential SVs for agronomic traits and domestication. The fact that this study lacks of novelty as raised by other reviewers is objective, nevertheless, it is also obviously of great importance for cucumber breeders and research community. Following the review process, it is seen that the manuscript has been carefully revised according to the previous reviewers, most of the reviewers' concerns have been addressed in the revised manuscript. Therefore, I don't have many concerns, but I do have some suggestions as listed below.

1. the previous reviewer #2 have the concerns about using resequencing data to evaluate the SVs identified in the pan-genome. The authors haven't given enough explanation in the response letter about why they acknowledged 'the limitation of using resequencing data to identify SVs in the introduction, but later used this method in the analysis.

Response: We are sorry about the confusion. The limitation of using resequencing data to *de novo* identify SVs has been widely admitted. However, in this study, we mainly utilized resequencing data to "genotype" SVs based on high-quality reference SVs identified from the graph-based pan-genome. Therefore, for a given SV, we checked if patterns of short-read mapping (split-read, read pairs) or sequencing coverages around the SV breakpoints were concordant with the type and length of this SV. In this case, we only leveraged the resequencing reads and relevant algorithms to "examine" if a given SV is truly present, rather than "calling" it.

2. line 317: I don't think the identification of SVs within CDS and putative promoter regions that associated with the change of expression of the closest gene can be evidence of concluding functions during domestication.

Response: We agree with the reviewer and have removed this description in this sentence (**Line 318**).

3. line 377-382, as suggested by reviewer#3, the authors have added the comparison of their results of structural variations and pan-genome with other plant species, however, the description is bland and unconstructive.

Response: We have revised the description regarding the comparison of SVs and removed the paragraph comparing the pan-genome results with other plant species, since no constructive information could be provided (**Line 378-381**).

4. line 387 the number of SVs in cucumber (53912) is not fewer than Medicago (27000-), at least not all the Medicago accessions, the statement must be concise and accurate.

Response: Sorry for the confusion. To achieve a fair comparison, we have re-calculated the number of SVs in cucumber and rice (large insertions and deletions, > 50 bp in size) and revised the corresponding description (**Line 378-381**).

5. Supplementary Fig. 3, would you explain why the interaction signal is significantly low in somewhere chr2 in all the three accessions?

Response: Telomeric and centromeric regions of cucumber are largely made up of highly repetitive satellite sequences that can span several mega base pairs⁷⁻⁹, which cannot be fully resolved even by long-read sequencing technologies. The highly repetitive nature of these sequences causes multiple-mapping issue of short Hi-C reads. Therefore, these regions exhibit much less contact signal in the heat map, for only uniquely mapped Hi-C reads were used to compute the interaction intensity (**Supplementary Fig. 3**).

Reviewer #5 (Remarks to the Author):

In this study, Li et al constructed a “pan-genome” of 12 cucumber lines, representing both wild and domesticated accessions. The authors put special emphasis on the identification of structural variations – both from the evolutionary perspective and explored the phenotypic consequences of few of these. The manuscript is written generally clearly and presents a large body of work – including the combination of several sequencing technologies to construct chromosome-scale assemblies of these 12 lines. I believe the study and resource will be of interest to researchers outside the immediate field and should span those interested in genomics and plant domestication. Still, I do not agree with some of the interpretations, some are quite central – specifically all those related to the evolutionary pathways proposed – and I suggest the authors carefully reexamine these sections – please see my detailed comments below (General comments #2 & #3, specific comments).

General comments:

1. The term “pan-genome” has been used rather loosely in the literature, from the analysis of very few accessions to hundreds, and as such this study is certainly on the small side. Based on simulation of pan-genome size, the authors argue the number of pan-genes reached a plateau when as low as nine accessions are used. Looking at Figure S5, it doesn't seem as if a plateau was reached. However, this also depends on how a plateau is defined, which was not defined by the authors. Arguing that the plateau was reached based on the plot is problematic since this depends on the resolution of the axes. As an alternative, it would be informative to specify the number of new genes that were added as more samples were included.

Response: We thank the reviewer for the suggestion. We computed the number of new gene clusters that were added as increasing accessions were included, and observed that it declined rapidly with only 98 new clusters when the number of accessions was 12 (**Response Fig. 1**). This result has been integrated into the main text (**Line 162-165**). This and other evaluations (Please check our response to question #1 of Reviewer #1) all supported that the 12 accessions used in pan-genome construction captured the majority of the genetic diversity in cucumber.

Response Fig. 1. Number of gene clusters that are newly added when including more accessions.

2. Karyotype evolution of cucumber (lines 148-159): I do not agree with this interpretation. Based on the phylogeny of figure 1, we can deduce that the inversion on chromosomes 4 & 7 occurred after divergence between W8 and Cuc64, and along the lineage leading to Cuc64. However, the phylogenetic location of the inversion on chromosome 5 is not clear, since the ancestral lineage could have possessed the karyotype of W8 or that of 9930 – in both cases, a single transition in either of the two ancestral lineages could lead to the observed data. Thus, the interpretation presented in lines 151-159 is one of two equally-parsimonious scenarios.

Response: Thanks for the comments! We rephrased this section to interpret karyotype evolution of cucumber more accurately (**Line 143-147**). “Based on the phylogeny of the 12 accessions (**Fig. 1d**), we can deduce that the inversions on chromosomes 4 and 7 could have occurred after the divergence between W8 and Cuc64. Considering that all cultivated and some wild accessions do not possess any inversions on chromosome 5, we propose that these inversions could have occurred during the evolution of wild cucumbers. The breakpoints and presence/absence information of these seven

chromosomal rearrangements among the 12 accessions provide novel insights into karyotype evolution in the *Cucumis* genus. Since large segmental inversions can lead to recombination suppression in these regions ², our inversion map of the 12 cucumber accessions provides a guide for properly selecting parental lines to construct segregating populations between wild and cultivated cucumbers.”

3. The analyses presented in figure 4b (and in lines 246-248) should be performed on the same phylogeny that is presented in figure 1 – why basing the analysis on a phylogeny derived from the small number of SNPs contained within these 6 genes? The same is true for the tree (and the corresponding analysis) presented in figure 6c.

Response: The reason that we chose the local phylogeny rather than the whole-genome phylogeny for the analysis is the widely existing incongruence between gene trees and species trees ¹⁰. For example, regarding the fruit spine/wart trait, the accession “9110gt” resembles the East-Asian cucumbers with high density of spines, which was supported by the phylogeny built using SNPs within the six genes: 9110gt and XTMC formed a monophyletic group (**Fig. 4a**). However, in the species tree constructed by the genome-wide single-copy ortholog genes, 9110gt was sister to the putative ancestor of the three East-Asian cucumbers (9930, XTMC and Cu2) (**Fig. 1d**). We think it would be more appropriate to display the “true” phylogeny reflecting the specific phenotype focused here.

For the results presented in **Fig. 6**, we divided the 12 accessions into three haplotypes based on the genotypes of the two identified SVs. Therefore, the “cladogram” shown in **Fig. 6c** was not a “real” phylogenetic tree but just an illustration of their evolutionary relationships considering ancestral states of these SVs.

4. It is unclear whether the analysis can distinguish translocations from large insertions. Can the authors comment on that?

Response: For large insertions, we first extracted syntenic regions between two genomes and perform pair-wise alignments to identify large insertions as well as large deletions. We applied SyRI ¹¹ to extracted non-syntenic rearranged regions and classified them into translocations and inversions based on their aligned conformation. This has been described in our original manuscript (**Line 641-651** and **Line 670-676**).

Specific comments:

5. Introduction. I don't think most readers are aware of the term "graph-based pan-genome". A more detailed explanation should help here.

Response: In the second paragraph of the introduction (**Line 71-73**), we have changed "thus emphasizing the need to build a species-wide graph-based pan-genome from diverse accessions" to "thus emphasizing the necessity to assemble additional reference genomes from more diverse accessions". In **Line 76-80**, we have rephrased these sentences to "In this study, we constructed chromosome-scale assemblies for an additional of 11 representative accessions comprising three wild and eight cultivated cucumbers. Together with the reference genome of 9930, we built a graph-based pan-genome by integrating them into a graph-like representation, within which each path referred to a possible sequence from one or more accessions". This should enable a better understanding to the readers.

6. Lines 69-71. Rephrase this sentence.

Response: We have changed this sentence from "Previous work also reported a resequencing-based SV map of 115 diverse accessions and a copy number variation (CNV) that defines the *Female (F)* locus, highlighting the role of SVs in favorable trait determination" to "A previously reported resequencing-based cucumber SV map revealed a copy number variation that defines the *Female (F)* locus, highlighting the role of SVs in favorable trait determination" (**Line 67-69**).

7. Line 77. How does the quality of the genome of line 9930 compares to the genomes produced here?

Response: The chromosome-scale genome of the 9930 reference line presents a comparable assembly completeness (total length of 224.8 Mb and BUSCO score of 97.7%) and a higher continuity (contig N50 of 8.9 Mb). We have integrated corresponding information of assembly and annotation of the 9930 genome into **Table 1**. Please see Li et al. ¹² for more details.

8. Line 79 . For the 11 representative accessions – should mention here how many wilds and how many cultivated lines.

Response: We have rephrased this sentence to “In this study, we constructed chromosome-scale assemblies for an additional of 11 representative accessions comprising three wild and eight cultivated cucumbers” (**Line 76-78**).

9. First page of the results - regarding the assembly strategy. The authors integrated several sequencing techniques to assemble these 11 lines, but the details are given as bits and pieces. Similarly, the coverage and genome quality statistics are separated into the different platforms (lines 99-100, 107-108, 110, 115) which is quite confusing. A paragraph that presents the overall assembly strategy could help the readers follow what and why was done.

Response: We have substantially re-written this section to achieve a better logic. Please see the revised manuscript from **Line 96 to Line 115**.

10. Line 117 - could be helpful to report the % complete BUSCOs on genome assemblies, not just genome annotations. Also – what was the BUSCO score of the reference 9930 genome?

Response: We have added one column describing the percentage of complete BUSCOs on the 11 genome assemblies in **Table 1** and also corresponding description in the main text (**Line 114**). The BUSCO score of the 9930 genome is 97.7% for assembly and 95.5% for gene annotation. The number of complete BUSCOs for gene annotation was slightly different from that for genome assembly, possibly due to different gene prediction strategies used in BUSCO and in this study. Corresponding information of assembly and annotation of the 9930 genome has also been added in **Table 1**. It is worth noting that the BUSCO scores have been updated based on the new version (v5.2.1) of the software and the newly released database embryophyta_odb10. We have also modified the technical details of BUSCO in the **Methods** section (**Line 553-555 and Line 567-568**).

11. Line 121 – what is the CDS length of these genes?

Response: The average length of CDS of these predicted genes ranged from 1,075 bp to 1,124 bp. We have added this information to the revised manuscript (**Line 124-125**).

12. Lines 125-136. Aside from this section, no analyses were performed with regards to transposable elements. Wither develop this analysis more thoroughly or else - I think this paragraph could be substantially shortened and appended to the previous paragraph.

Response: Thanks for the suggestion. We have removed some less important description in this paragraph and integrated it into the previous paragraph (**Line 116-128**).

13. Lines 144-145. This sentence is unclear. What does it mean “discrete chromatin interactions around their breakpoints”?

Response: We have rephrased this sentence to “Mapping Hi-C data of Cuc64 to the genome of 9930 reference showed strong interaction signals around breakpoints of all these inversions” (**Line 136-137**).

14. Line 180. The KaKs analysis does not suggests that dispensable genes have undergone greater positive selection, but that dispensable genes have undergone less stringent purifying selection.

Response: We have changed “greater positive selection” to “less stringent purifying selection” (**Line 175**). Thanks.

15. Lines 196-197. The definition of complex InDels is confusing. It would be helpful to better describe these and what type of actual mutation they may represent.

Response: We have changed the sentence “complex InDels, which were the situation that one sequence (> 1 bp in size) was replaced by another one with different length without a precise breakpoint boundary” to “complex InDels that are pairs of un-aligned sequences of different lengths within regions in good collinearity” in **Line 191-192** of the revised manuscript.

16. Line 191 - how robust are the results to the choice of 'pivot' genome? What if you chose another one rather than the reference? What would be the effect on the catalog of SVs?

Response: Since the genome of line 9930 has the highest continuity (contig N50 of 8.9 Mb) and similar completeness (total length of 224.8 Mb and BUSCO score of 97.7%)

among the 12 cucumber accessions used here, we chose the 9930 genome as the “pivot” reference for variant identification. Considering that all the 11 genomes and the genome of line 9930 assembled in this study were built using PacBio long reads and presented similar completeness (**Table 1**), we do not expect notable changes if switching to another reference genome for identifying variants. Regarding catalog of SVs, for a specific case that an insertion present in the Cuc64 genome relative to 9930 (here the 9930 genome is used as the reference): if we select the Cuc64 genome as the reference, this insertion will be “converted” to a deletion based on our pipeline, while inversions and translocations will not be affected except only the adjustment of coordinates of SV breakpoints on different reference genomes.

17. Line 214 - can the catalog of SVs and/or the SV graph be used to detect gene PAV? It could be interesting to compare this to the results obtained in the previous section by annotation and clustering of genes.

Response: To examine the possibility of utilizing SVs to identify PAVs of protein-coding genes, we took the large insertions as an example. We first extracted sequences of large insertions from each of the 11 accessions compared with the 9930 reference and removed inserted fragments displaying high levels (>95%) of identity with the 9930 genome. The strategy applied here was similar to Thir et al ¹³. The threshold (95%) defined here was the same as the identity cutoff when performing gene clustering using protein sequences. This resulted in 0.88 - 6.33 Mb of non-reference inserted segments in each of the 11 accessions relative to the 9930 genome (presence variants, PVs). We then retrieved 809 genes within these PVs and compared them with the 2,354 pan-gene clusters in which genes from the 9930 genome were absent. Nearly 90% (718 out of the 809) of them were also detected in those clusters, while genes within the remaining 1,636 clusters could not be identified via the SV-based approach. Several reasons could lead to this: 1) Genomic regions that did not contain SVs could also encode genes with extensive dissimilarity in their amino-acid sequences due to the presence of, e.g., frameshift InDels or variants leading to gain or loss of stop codons. 2) Gene prediction pipelines might not annotate “all” the gene models in a genome, which introduced errors into gene clustering. We have partially resolved this issue through “rescuing” potentially unpredicted genes by aligning annotated CDS from other genomes to a given genome (**Methods, Line 562-567**). However, the first issue cannot

be addressed by merely considering SVs, which will require more efforts on analyzing increasing types of genetic variants. Thus, we think the gene clustering-based method applied in this study should capture the majority of gene PAV, and is more straightforward and efficient than the SV-based strategy.

18. Line 219. It is unclear against what phenotype the GWAS was performed. I see that this information is presented in the Supp Materials, but it should also be noted in the main text.

Response: We have added corresponding description on traits that were analyzed in GWAS and also revised this section for a better logic (**Line 214-217**).

19. Lines 293-295. It is not clear why lines CG0001 and CG0002 are considered as the ancestral lines – the ancestor types could have been W4 just as well.

Response: We have changed the sentence “These two accessions were considered to be the putative ancestor of all other cucumbers according to phylogeny of the 115 cucumber lines” to “These two accessions belong to the wild form of cucumber (*C. sativus* var. *hardwickii*) from which cultivated cucumbers were putatively domesticated¹⁴” (**Line 290-292**)

20. Line 308. Change to “located in regions that have undergone domestication sweeps”

Response: We have revised this sentence according to the reviewer’s comment to “located in regions that have undergone selection during cucumber domestication” (**Line 304-305**).

21. Line 317. “131 ARE within CDS regions and 1,480 ARE within...”

Response: We have changed the sentence “Of these dSVs and pdSVs, 131 within CDS regions and 1,480 within putative promoter regions (2-kb upstream of genes)” to “We found that 1,611 of these dSVs and hdSVs, of which 131 are within CDS regions and 1,480 are within putative promoter regions” (**Line 314-316**). Note that we have also changed “pdSVs” to “hdSVs (highly divergent SVs)” based on comments from Reviewer #2.

22. Line 366: “evolutionary processes”

Response: We have changed the word “evolution” to “evolutionary” (**Line 363**).

23. Line 371: “these studies may HAVE overlookED”

Response: We have fixed the grammar in the revised manuscript (**Line 368-369**).

24. Lines 379-381. The proportion of core genes greatly depends on the exact definition of core genes used in each study and on the number of accessions used in each pan-genome. Thus, this comparison should be re-made while using the same definition for all pan-genomes. For example, the percentage reported for brachipodium of 54% includes also genes that are missing from one accession.

Response: We have removed the paragraph comparing the pan-genome results with other plant species, since no constructive information could be provided.

25. Line 395 - it is quite possible that the low genomic diversity observed in cucumbers is the result of the methods applied in this study and the low number of samples - this (along with other limitations of the method) should be discussed here.

Response: We have added two sentences discussing the limitation of this study. One is “However, it is also possible that the relatively small sample size in this study could also lead to the underestimation of the biodiversity. Nonetheless, the methodology applied in this research presents a road map for further studies to characterize the full spectrum of genetic diversity by assembling genomes from more accessions within this important vegetable crop” in **Line 386-390**. The other is “Nevertheless, our study did not incorporate SNPs and small InDels into the graph-based pan-genome, which will be worth further investigation on how the graph structure can improve calling or genotyping of these variants” in **Line 397-399**.

26. Figure 3 – I find it confusing that panels (d) and (e) are displayed as heat maps whereas (f) and (g) are displayed as line plots, while they describe similar things.

Response: We have changed panels **d** and **e** to line plots, similar as panels **f** and **g** to avoid confusion.

27. Figure 3 – I found panels (h) and (i) quite confusing. Should better explain what is shown.

Response: We have added more detailed information in both panels **h** and **i** and figure legends.

28. Figure 4c - the IGV screenshot will probably be difficult to understand for readers who don't regularly use this software. I don't think showing this plot is necessary, but for sure another way to display this is needed.

Response: We have changed this plot to a genome coverage line plot around this SV to make it easier to understand. Legend of **Fig. 4c** has also been changed.

Reference

1. De Beukelaer, H., Davenport, G.F. & Fack, V. Core Hunter 3: flexible core subset selection. *BMC Bioinformatics* **19**, 203 (2018).
2. Huang, S. *et al.* The genome of the cucumber, *Cucumis sativus* L. *Nat. Genet.* **41**, 1275-81 (2009).
3. Zhang, H. *et al.* A fragment substitution in the promoter of *CsHDZIV11/CsGL3* is responsible for fruit spine density in cucumber (*Cucumis sativus* L.). *Theor. Appl. Genet.* **129**, 1289-301 (2016).
4. Yang, X. *et al.* Tuberculate fruit gene *Tu* encodes a C₂H₂ zinc finger protein that is required for the warty fruit phenotype in cucumber (*Cucumis sativus* L.). *Plant J.* **78**, 1034-1046 (2014).
5. Zhang, Z. *et al.* Genome-wide mapping of structural variations reveals a copy number variant that determines reproductive morphology in cucumber. *Plant Cell* **27**, 1595-604 (2015).
6. Simao, F.A., Waterhouse, R.M., Ioannidis, P., Kriventseva, E.V. & Zdobnov, E.M. BUSCO: assessing genome assembly and annotation completeness with single-copy orthologs. *Bioinformatics* **31**, 3210-2 (2015).
7. Ganall, M. & Hemleben, R. Organization and sequence analysis of two related satellite DNAs in cucumber (*Cucumis sativus* L.). *J. Mol. Evol.* **23**, 23-30 (1986).
8. Ganall, M. & Hemleben, R. Insertion and amplification of a DNA sequence in satellite DNA of *Cucumis sativus* L. (cucumber). *Theor. Appl. Genet.* **75**, 357-361 (1988).
9. Han, Y.H. *et al.* Distribution of the tandem repeat sequences and karyotyping in cucumber (*Cucumis sativus* L.) by fluorescence *in situ* hybridization. *Cytogenet. Genome Res.* **122**, 80-8 (2008).
10. Wu, Y. Coalescent-based species tree inference from gene tree topologies under incomplete lineage sorting by maximum likelihood. *Evolution* **66**, 763-775 (2012).
11. Goel, M., Sun, H., Jiao, W.B. & Schneeberger, K. SyRI: finding genomic rearrangements and local sequence differences from whole-genome assemblies. *Genome Biol.* **20**, 277 (2019).
12. Li, Q. *et al.* A chromosome-scale genome assembly of cucumber (*Cucumis sativus* L.). *Gigascience* **8**(2019).
13. Tahir Ul Qamar, M., Zhu, X., Xing, F. & Chen, L.-L. ppsPCP: a plant presence/absence variants scanner and pan-genome construction pipeline.

Bioinformatics **35**, 4156-4158 (2019).

14. Qi, J. *et al.* A genomic variation map provides insights into the genetic basis of cucumber domestication and diversity. *Nat. Genet.* **45**, 1510-5 (2013).

Reviewers' Comments:

Reviewer #1:

Remarks to the Author:

The authors have gone to great lengths to address my concerns and those of the other reviewers. The manuscript has been significantly improved. Yet, it pains me to say so, but I am still not entirely convinced that the data represented in the revised manuscript supports the statement that the 12 selected accessions represent a pan-genome. In response to this original concern of mine the authors extracted 10,000 random SNPs (with missing call rates of less than 0.1) from the WHOLE population of 115 accessions and proceeded to compute various diversity indices. They argue that these indices suggest a pan-genome, however, the results are not reported in the revised manuscript. Instead, the authors conduct (and report in the revised manuscript) a diminishing return experiment on the SUB-POPULATION of 12 sequenced accessions to show that the number of new gene clusters detected decays as more accessions are analyzed. I would expect this to be the case when analyzing a sub-population. The authors did not establish what proportion of the total diversity the 12 accessions represent. The reader is left to wonder if this is 70%, 80%, 90% or 95%... The pan-genome statement should be supported entirely by the data represented in the manuscript, as this is how the paper will be presented to the community upon publication.

It would appear that Reviewer 5 also had similar concerns - indeed, this was the first and foremost concern raised by this reviewer. My disquietude would be dampened if Reviewer 5 is satisfied that the additional analysis reported in the revised manuscript now supports the threshold of a pan-genome.

Apart from this, I am happy with how the authors have addressed the remaining concerns that I had.

Reviewer #2:

Remarks to the Author:

The authors have addressed all of my previous comments and concerns and I am now happy with the manuscript.

Reviewer #3:

Remarks to the Author:

The authors have adequately addressed my previous questions. I do not have further questions.

Reviewer #4:

Remarks to the Author:

The authors have done many efforts to improve the manuscript, all my previous concerns were adequately addressed. I don't have any more questions.

Reviewer #5:

Remarks to the Author:

The authors have addressed adequately most of my comments - specifically those related to the evolutionary interpretation.

As a minor comment, I still think that the use of the buzz-term "graph-based genome" is not well explained to readers who are not acquainted with this term.

Reviewer notes:**Reviewer #1 (Remarks to the Author):**

The authors have gone to great lengths to address my concerns and those of the other reviewers. The manuscript has been significantly improved. Yet, it pains me to say so, but I am still not entirely convinced that the data represented in the revised manuscript supports the statement that the 12 selected accessions represent a pan-genome. In response to this original concern of mine the authors extracted 10,000 random SNPs (with missing call rates of less than 0.1) from the WHOLE population of 115 accessions and proceeded to compute various diversity indices. They argue that these indices suggest a pan-genome, however, the results are not reported in the revised manuscript. Instead, the authors conduct (and report in the revised manuscript) a diminishing return experiment on the SUB-POPULATION of 12 sequenced accessions to show that the number of new gene clusters detected decays as more accessions are analyzed. I would expect this to be the case when analyzing a sub-population. The authors did not establish what proportion of the total diversity the 12 accessions represent. The reader is left to wonder if this is 70%, 80%, 90% or 95%... The pan-genome statement should be supported entirely by the data represented in the manuscript, as this is how the paper will be presented to the community upon publication.

It would appear that Reviewer 5 also had similar concerns - indeed, this was the first and foremost concern raised by this reviewer. My disquietude would be dampened if Reviewer 5 is satisfied that the additional analysis reported in the revised manuscript now supports the threshold of a pan-genome.

Apart from this, I am happy with how the authors have addressed the remaining concerns that I had.

Response: We thank the reviewer for raising this concern. To address this, in the revised manuscript we have added the comparison of genetic distance measures and diversity index between the 12 accessions used in this study and the 115 lines from the core collection. We have also calculated genetic coverage value using 10,000 randomly selected SNPs with 20 independent replicates and found that ~84% of the genetic diversity of the 115-line core collection was captured by the 12 accessions, indicating the representativeness of the 12 accessions used in pan-genome construction. These results have been added in the revised manuscript (**Line 84-92** and **Supplementary Table 1**). The relevant methods have also been added in the revised manuscript (**Line 436-442**).

Reviewer #2 (Remarks to the Author):

The authors have addressed all of my previous comments and concerns and I am now happy with the manuscript.

Response: Thanks.

Reviewer #3 (Remarks to the Author):

The authors have adequately addressed my previous questions. I do not have further questions.

Response: Thanks.

Reviewer #4 (Remarks to the Author):

The authors have done many efforts to improve the manuscript, all my previous concerns were adequately addressed. I don't have any more questions.

Response: Thanks.

Reviewer #5 (Remarks to the Author):

The authors have addressed adequately most of my comments - specifically those related to the evolutionary interpretation.

As a minor comment, I still think that the use of the buzz-term "graph-based genome" is not well explained to readers who are not acquainted with this term.

Response: We thank the reviewer for pointing this out. We have rephrased corresponding description of "graph-based genome" as well as the legend of **Fig. 3h** to make it clear (**Line 54-57 and Line 188-193**).

Reviewers' Comments:

Reviewer #1:

Remarks to the Author:

With the additional analyses the authors have now addressed my concerns about the selected accessions as representative of the cucumber pan-genome. I have no further reservations.

Reviewer notes:

Reviewer #1 (Remarks to the Author):

With the additional analyses the authors have now addressed my concerns about the selected accessions as representative of the cucumber pan-genome. I have no further reservations.

Response: Thanks.